GENETICS, 2024, iyae003

**Multiparental Populations**

# Characterization of adaptation mechanisms in sorghum using a multireference back-cross nested association mapping design and envirotyping

Vincent Garin [ID],[1,2,3,†] Chiaka Diallo [ID],[4,5,†] Mohamed Lamine Tékété [ID],[6,7,†] Korotimi Théra [ID],[6,†] Baptiste Guitton,[2,3] Karim Dagno,[6] Abdoulaye G. Diallo,[6] Mamoutou Kouressy,[6] Willmar Leiser [ID],[4] Fred Rattunde,[8] Ibrahima Sissoko,[4] Aboubacar Touré,[4] Baloua Nébié [ID],[9] Moussa Samaké,[7] Jana Kholovà,[1,10] Angélique Berger,[2,3] Julien Frouin [ID],[2,3] David Pot [ID],[2,3] Michel Vaksmann [ID],[2,3] Eva Weltzien,[4,8,*] Niaba Témé,[6,*] Jean-François Rami [ID][2,3,*]

[1]Crop Physiology Laboratory, International Crops Research Institute for the Semi-Arid Tropics, Patancheru, 502 324, India
[2]CIRAD, UMR AGAP Institut, Montpellier, F-34398, France
[3]UMR AGAP Institut, Univ Montpellier, CIRAD, INRAE, Institut Agro, Montpellier, F-34398, France
[4]Sorghum Program, International Crops Research Institute for the Semi-Arid Tropics, Bamako, BP 320, Mali
[5]Département d'Enseignement et de Recherche des Sciences et Techniques Agricoles, Institut polytechnique rural de formation et de recherche appliquée de Katibougou, Koulikoro, BP 06, Mali
[6]Institut d'Economie Rurale, Bamako, BP 262, Mali
[7]Faculté des Sciences et Techniques, Université des Sciences des Techniques et des Technologies de Bamako, Bamako, BP E 3206, Mali
[8]Agronomy Department, University of Wisconsin, Madison, WI 53705, WI, USA
[9]Dryland Crops Program, International Maize and Wheat Improvement Center (CIMMYT-Senegal) U/C CERAAS, Thiès, Po Box 3320, Senegal
[10]Department of Information Technologies, Faculty of Economics and Management, Czech University of Life Sciences, Prague, 165 00, Czech Republic

*Corresponding author: UMR AGAP Institut, CIRAD, University of Montpellier, Avenue d'Agropolis, Montpellier, F-34398, France. Email: jean-francois.rami@cirad.fr;
*Corresponding author: Laboratoire de biotechnologie, Institut d'Economie Rurale, Rue Mohamed V, Bamako, BP 258, Mali. Email: niabateme@gmail.com;
*Corresponding author: Agronomy Department, University of Wisconsin, 371 Moore Hall, 1575 Linden Drive, Madison, WI 53705, USA. Email: eva.weltzien@gmail.com
[†]C.D., V.G., M.L.T., and K.T. contributed equally to this work.

Identifying the genetic factors impacting the adaptation of crops to environmental conditions is of key interest for conservation and selection purposes. It can be achieved using population genomics, and evolutionary or quantitative genetics. Here we present a sorghum multireference back-cross nested association mapping population composed of 3,901 lines produced by crossing 24 diverse parents to 3 elite parents from West and Central Africa-back-cross nested association mapping. The population was phenotyped in environments characterized by differences in photoperiod, rainfall pattern, temperature levels, and soil fertility. To integrate the multiparental and multi-environmental dimension of our data we proposed a new approach for quantitative trait loci (QTL) detection and parental effect estimation. We extended our model to estimate QTL effect sensitivity to environmental covariates, which facilitated the integration of envirotyping data. Our models allowed spatial projections of the QTL effects in agro-ecologies of interest. We utilized this strategy to analyze the genetic architecture of flowering time and plant height, which represents key adaptation mechanisms in environments like West Africa. Our results allowed a better characterization of well-known genomic regions influencing flowering time concerning their response to photoperiod with *Ma6* and *Ma1* being photoperiod-sensitive and the region of possible candidate gene *Elf3* being photoperiod-insensitive. We also accessed a better understanding of plant height genetic determinism with the combined effects of phenology-dependent (*Ma6*) and independent (*qHT7.1* and *Dw3*) genomic regions. Therefore, we argue that the West and Central Africa-back-cross nested association mapping and the presented analytical approach constitute unique resources to better understand adaptation in sorghum with direct application to develop climate-smart varieties.

Keywords: multiparental populations; quantitative trait loci; multireference BCNAM; genotype by environment interaction; envirotyping; adaptation

## Introduction

The genotype by environment (GxE) interaction, which manifests by differential genotypic expressions across environmental conditions (Falconer and Mackay 1996), is an important phenomenon impacting plant development in natural and controlled environments (Campbell and Waser 2001; Annicchiarico 2002). GxE is a fundamental component of plant adaptation (Des Marais *et al.* 2013), which strongly influences the expression of complex traits (van Eeuwijk *et al.* 2010). For example, the difference in development response to the length of the day (photoperiod) is a remarkable example of GxE interaction (Nakamichi 2015). An important proportion of genes shows environment-specific behavior, which considerably complicates their use for selection (Bernardo 2016; Cobb *et al.* 2019). Therefore, the determination of the proportion

of genes influenced by the environment as well as the nature of this influence is a central question. The availability of complex genetic resources, advanced technology to monitor plants in different environmental conditions (envirotyping; Xu 2016), and accurate statistical methods should improve our understanding of the GxE interaction in plants.

Multiparental populations (MPPs) combining the genomes of several founders like the nested association mapping (NAM) design or the multiparent advanced generation inter-cross (MAGIC) design have progressively emerged as central genetic resources for research (Cavanagh *et al.* 2008; Scott *et al.* 2020; Bernardo 2021). The NAM design composed of crosses between a recurrent parent and donor parents is a well-spread MPP design (Yu *et al.* 2008; McMullen *et al.* 2009; Gage *et al.* 2020) with examples in maize (Bauer *et al.* 2013; Chen *et al.* 2019), rice (Fragoso *et al.* 2017), wheat (Kidane *et al.* 2019; Altendorf *et al.* 2021; Christopher *et al.* 2021), barley (Maurer *et al.* 2015; Hemshrot *et al.* 2019) and sorghum (Bouchet *et al.* 2017; Marla *et al.* 2019). Sorghum is also the species that was used to develop the back-cross NAM (BCNAM) design, which consists of introgressing diverse alleles from donors in a recurrent (elite) line using 1 generation of back-cross followed by several generations of selfing (Jordan *et al.* 2011; Mace *et al.* 2021).

The BCNAM design has several interesting properties for genetic analyses. Compared to biparental crosses, it addresses a larger genetic diversity and captures more recombination events. Compared to association panels, it offers better control over the population structure, which can reduce false-positive detection (Myles *et al.* 2009). BCNAM designs also allow the origin of favorable alleles to be traced back to a specific parent, a highly desirable feature in the design of future crosses. MPPs like BCNAM increase rare allele frequencies, which is essential to the precise estimation of their additive effects (Myles *et al.* 2009). Moreover, an extension of the reference NAM design by the use of several recurrent parents allows the characterization of the genetic effect in multiple genetic backgrounds (Christopher *et al.* 2021). BCNAM designs are also interesting in the study of GxE because they allow the exposure of structured diversity to contrasting environments, which can maximize the detectable allelic expression (Cobb *et al.* 2019).

Several approaches have been developed for the detection of single-environment MPP quantitative trait loci (QTL). For example, Li *et al.* (2011) developed a method based on maximum likelihood QTL effect significance. Xavier *et al.* (2015) used mixed models employed for genome-wide association analysis. Garin *et al.* (2017, 2018) proposed a framework assuming different allelic configurations at the QTL position. More recently, Paccapelo *et al.* (2022) adapted the whole-genome interval mapping method for the NAM design. A more general strategy consists of using models based on probabilistic estimation of identity by descent (IBD) calculated in any type of design (Zheng *et al.* 2015; Broman 2022). This information can be integrated into models using random QTL effects with the IBD matrix associated (Wei and Xu 2016; Li *et al.* 2021). A last solution is to estimate MPP QTL effects with a Bayesian approach, which offers an elegant solution to the question of model determination (Wu and Jannink 2004). By treating more sources of randomness in the estimation procedure, can also offer greater precision for haplotype or allelic series effect estimation, but at the price of a higher computational demand (Zhang *et al.* 2014; Crouse *et al.* 2020).

Compared to separate within-environment analyses, the QTL detection using MPP data characterized in multiple environments (MPP-ME) in a joint model is more challenging, but it allows a more direct comparison of the estimated effects. Until now, phenotypic values were averaged across environments (e.g. Giraud *et al.* 2014), which prevents the use of the full potential of those data like the

environmental correlation (Piepho and Pillen 2004). To overcome this limitation, Verbyla *et al.* (2014) proposed a QTL detection approach for MAGIC populations characterized in ME that allows estimation of environment-specific parental QTL effects. In the same vein, Garin *et al.* (2020) extended their framework to estimate MPP-ME QTL models with parental or ancestral haplotype effects. Diouf *et al.* (2020) proposed a forward–backward algorithm for MPP-ME analysis in MAGIC. Finally, De Walsche *et al.* (2023) proposed approaching MPP-ME QTL detection as a meta-analysis of single environment analyses.

The extension of multi-environment QTL models with environmental covariates (ECs) for biparental populations and MPPs has proven useful in estimating the sensitivity of the QTL effect to specific dimensions of the environment (Malosetti *et al.* 2004; Boer *et al.* 2007; Garin *et al.* 2020). Recent advances in sensor technology have considerably increased the availability of high-resolution environmental characterization and the possibility of integrating this information into genetic analysis, often termed envirotyping (Xu 2016; Costa-Neto *et al.* 2021). In this study, we propose an approach to integrate the available high-resolution environmental information in MPP-ME QTL models to improve our understanding of the QTLxE interaction by testing the sensitivity of multiple alleles to various ECs.

Photoperiod is a key EC for sorghum development, especially in West Africa. Photoperiodism is the developmental response of plants to the relative length of daylight or photoperiod (Hopkins 2008). The influence of photoperiod on flowering time is particularly important for sorghum because it represents an important source of adaptation by adjusting the length of the vegetative and reproductive phases given local environmental conditions (Kouressy *et al.* 2008). Sorghum is a short-day plant generally sensitive to photoperiod that delays its panicle initiation when days become longer (Wolabu and Tadege 2016). The flowering time can be represented as a broken linear function of the photoperiod with a baseline duration remaining constant until a certain photoperiod and then an increasing slope where flowering time increases with the photoperiod (Van Oosterom *et al.* 2001; Fig. 4g). Photoperiod sensitivity is the variation in the steepness of the slope. Next to photoperiod, temperature is another environmental covariate that is assumed to influence sorghum flowering time at the genetic level (Mace *et al.* 2013).

In this article, we present a multireference sorghum BCNAM of populations composed of 24 diverse parents anchored on 3 West African elite lines that represents one of the most relevant publicly available resources for West and Central Africa sorghum (WCA-BCNAM, Table 1, Supplementary Fig. 1). The population was phenotyped for 8 traits covering phenology and yield component aspects of the plant in ME. To analyze those data, we developed MPP-ME QTL models integrating environmental covariables. We illustrate our approach through a fine characterization of major QTL for flowering time and plant height.

## Material and methods

Figure 1 is an overview of the proposed analytical strategy. In the following sections, we will first describe the data used. Then, we will present the statistical models for phenotypic analyses, MPP QTLxE analysis extended with ECs, and the procedure to project QTL behavior beyond the tested environments.

### Plant material

The WCA-BCNAM is composed of 3 BCNAM populations produced after the crossing of the Grinkan (GR), Kenin-Keni (KK), and Lata3

**Table 1.** WCA-BCNAM parents with racial classification, origin, relative height (PH), relative maturity, reaction to photoperiod sensitivity (Kp3), and specific advantages. The last 3 columns specify the crossing scheme with the year when the cross was phenotyped (2012 and/or 2013).

| Parent information | | | | | | | | Crossing scheme | | |
|---|---|---|---|---|---|---|---|---|---|---|
| Parent | Race | Origin | PH | Mat | Kp3 | Specific advantage | References | GR | KK | LT |
| GR | G/C | Mali | av | av | av | Elite line | Guitton *et al.* (2018) | | | 13 |
| KK | G/C | Mali | av | av | av | Elite Line | Leroy *et al.* (2014) | | | |
| LT | G/C | Mali | + | av | av | Elite Line | Diallo *et al.* (2019) | | | |
| Fara-Fara | G | Nigeria | + | + | + | Diversity | Andrews (1973) | 12 | 12 | 13 |
| E36-1 | C | Ethiopia | − | av | − | Drought tolerance | Mahalakshmi and Bidinger (2002) | 12/13 | 12 | |
| IS15401 | G | Cameroon | + | + | + | Striga resistance | FAO (2008) | 12 | 12 | 13 |
| IS23540 | C | Ethiopia | − | av | − | Sweet stem | FAO (2023) | 12 | 13 | 13 |
| B35 | D/C | Ethiopia | − | − | − | Drought tolerance | Rama Reddy *et al.* (2014) | 12 | 12 | |
| Konotene | D | Mali | + | + | − | Grain weight | Clément *et al.* (1980) | 12 | | |
| SC566-14 | C | Nigeria | − | − | − | Al tolerance | Magalhaes *et al.* (2004) | 12 | | 13 |
| Framida | C | S. Africa | − | + | − | Striga resistance | Haussmann *et al.* (2001) | 12 | | 13 |
| CSM417 | G | Mali | + | + | + | Grain quality | Clément *et al.* (1980) | 12 | 12/13 | |
| CSM63 | G | Mali | av | − | − | Precocity | Chantereau *et al.* (1998) | 12 | | |
| CSM388 | G | Mali | + | + | + | Grain quality | Folliard *et al.* (2004) | 12/13 | | |
| Gadiaba Dié | D | Mali | + | + | + | Grain weight | Clément *et al.* (1980) | 12 | | |
| W. Kaura | D/C | Nigeria | − | + | + | Diversity | Goma *et al.* (2012) | 12 | | |
| V33/08 | G/C | Mali | av | + | av | Grain quality | Soumaré *et al.* (2008) | 13 | | |
| Kalaban | C | Mali | − | av | − | Productivity | FAO (2008) | 13 | 13 | |
| Malisor 84-7 | C | Mali | − | − | + | Head bug resistance | Ratnadass *et al.* (2002) | 13 | 13 | |
| BimbG | G | Guinea | + | + | + | Grain quality | Sagnard *et al.* (2011) | 13 | 13 | |
| Hafijeka | G | Gambia | + | + | + | Grain quality | Folkertsma *et al.* (2005) | 13 | | 13 |
| S. Kaura | D/C | Nigeria | + | + | + | Diversity | Kassam and Andrews (1975) | 13 | | 13 |
| Sangatigui | G | Mali | av | av | av | Diversity | CEDEAO-UEMOA-CILSS (2016) | | 13 | |
| DouaG | G | Mali | + | + | + | Low-P adaptation | Kante *et al.* (2017) | | | 13 |
| Gnossiconi | G | Burkina F. | av | av | av | Yield stability | vom Brocke *et al.* (2014) | | | 13 |
| Ngolofing | G | Mali | + | av | + | Grain quality | Clément *et al.* (1980) | | | 13 |
| Sambalma | G/C | Nigeria | + | + | + | Al tolerance | Kante *et al.* (2019) | | | 13 |

Race: D: durra; C: caudatum; G: guinea. Photoperiod sensitivity (Kp3):—Kp < 0.4; av 0.4 < Kp < 0.6; + 0.7 < Kp < 1. PH:—PH < 1 m; av 1m < PH < 2 m; + PH > 2 m. Maturity (cycle):—cycle <70 days; av 70 days <cycle < 110 days; +cycle >110 days (These estimations were carried out under early sowing conditions (June) in Mali; Thera 2017).

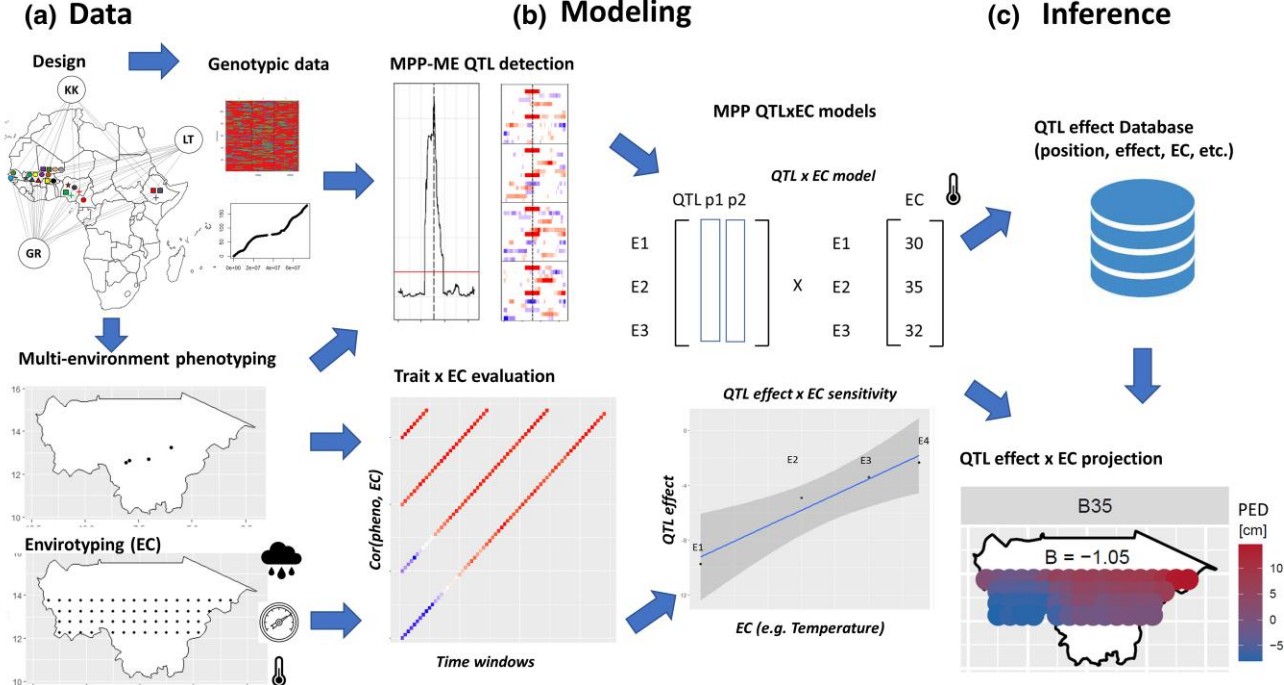

**Fig. 1.** Overview of the analytical strategy. a) Raw genotypic, phenotypic and environmental covariables; b) Statistical models for QTL detection in MPP characterized in ME, correlations between trait and EC analysis, and synthesis in QTLxEC models; c) Inference using the results gathered in a database and projection of the QTL effect beyond the tested environments.

(LT) recurrent parents with 24 donor parents representative of the Western African sorghum diversity with lines from Central and Eastern Africa (Table 1, Supplementary Table 1: parents name synonyms). The whole population contains 3,901 BC$_1$F$_4$ genotypes from 41 crosses (GR: 2,109 genotypes, 19 crosses; KK: 896 genotypes 10 crosses; LT: 896 genotypes, 12 crosses, Supplementary Fig. 1). The LT population crosses involved male sterile sister lines of LT to produce the BC$_1$ generation, while the BC$_1$ generations of GR and KK crosses were produced using manual emasculation. The recurrent parents are elite lines selected in Mali through farmer variety testing. GR was developed through pedigree breeding methods. KK was derived from a directed recurrent selection population involving local parents of different botanical types (Leroy *et al.* 2014). LT was selected from a random mating population of Guinea parents (Diallo *et al.* 2019). During the development of the populations, moderate selection pressure was applied at the BC$_1$F$_2$ generation against too early flowering and too high genotypes.

The recurrent parents were chosen for their productivity, their adaptation to soil and climate, and their resistance to major biotic and abiotic stresses. The recurrent lines also have weaknesses like poor grain quality and mold susceptibility (GR), suboptimal glume opening and/or susceptibility to *Striga hermonthica* (LT), and low productivity and yield stability (KK). The 24 donor parents cover diverse racial (Guinea, Caudatum, Durra) and geographical origins (Table 1). They are characterized by key adaptive traits like height, maturity, and photoperiod sensitivity (Kp3). Those parents were also selected for traits like tolerance to *Striga hermonthica*, soil phosphorus deficiency and/or drought, and good grain quality that could increase farmer acceptance. Several donor parents were tested with multiple genetic backgrounds. Fara-Fara, IS15401, and IS23540 were crossed with the 3 recurrent parents. Ten donor parents were tested in 2 genetic backgrounds.

## Genotypic data

The 3,901 offspring and their parents were genotyped using genotype by sequencing (GBS, Elshire *et al.* 2011) with 384-plex libraries on an Illumina HiSeq 2000 sequencer. The offspring were genotyped at generation BC$_1$F$_3$. The sequence data were analyzed running the reference genome-based TASSEL GBS pipeline (Glaubitz *et al.* 2014). Unique tags (3,844,911) were aligned on the sorghum reference genome v2.1 (Paterson *et al.* 2009). After the filtering of raw genotype data for minor allele frequencies (MAF < 0.05) and single marker missing data (<0.9), 51,545 segregating single nucleotide polymorphisms (SNPs) were identified between the parents with between 11,856 and 26,128 SNPs segregating in the individual crosses. Missing values in the parents were imputed using Beagle (Browning *et al.* 2018). Missing values in the offspring genotypes were imputed using FSFHap (Bradbury *et al.* 2007). We determined a unique genetic consensus map (Supplementary Table 2) by projecting the physical distance of the 51,545 markers on a high-quality genetic consensus map (Guindo *et al.* 2019) using the R package ziplinR (https://github.com/jframi/ziplinR).

## Phenotypic data

Because of logistic constraints, the GR and KK populations could not be entirely phenotyped during 1 season. Therefore, the crosses of those populations were phenotyped over 2 seasons (2012 and 2013, Table 1, Supplementary Fig. 1). Due to minimal overlap between the genotypes phenotyped in 2012 and 2013 (maximum of 2 crosses per recurrent parent), GR (GR2012, GR2013) and KK (KK2012, KK2013) subpopulations were analyzed independently. Each subpopulation was phenotyped in a combination of

2 locations (Sotuba and Cinzana, Fig. 5a) for 2 sowing dates (Sowing 1: end of June, sowing 2: 3–4 weeks later, Supplementary Table 3), totaling 4 environments (SB1, SB2, CZ1, CZ2; Supplementary Fig. 2). The Sotuba location is characterized by around 900 mm/year of precipitation and lower temperatures, while the Cinzana location is characterized by lower precipitation (600–700 mm/year) and higher temperatures. In both locations, the second sowing date had a lower level of precipitation and humidity and higher temperatures (Supplementary Fig. 2, Supplementary Table 4). In each environment, the progenies of the GR and KK populations were laid out as an augmented design (Kempton 1984), with the 3 recurrent parents used as checks. The LT population was entirely phenotyped in 2013 in 3 environments defined by 2 levels of phosphorus fertilization (low-P and HIGH-P) at the Samanko station and standard conditions at the Kolombada station (Fig. 5a). In each environment, the genotypes were laid out as an alpha-lattice design (Kempton 1984) with 2 replications (Diallo *et al.* 2019).

We measured 8 traits listed hereafter with crop ontology (CO) code. Flag leaf appearance (FLAG, CO_324:0000631) was the number of days after sowing when half of the plot had their ligulated flag leaves visible. For the QTL analysis, FLAG data were converted into degree day (DD). Plant height (PH, CO_324:0000623) was the distance in cm between the soil and the panicle top. The number of internodes (NODE_N, CO_324:0000605) was the number of nodes on the main stem −1 and the average length of the internodes (NODE_L) was the main stem length divided by NODE_N. The peduncle length (PED, CO_324:0000622) was the distance in cm between the final node and the panicle bottom. The panicle length (PAN, CO_324:0000620) was the distance in cm from the end of the peduncle to the panicle top. Grain weight (GWGH) (CO_324:0000424) was the weight in grams of 1000 grains. Finally, YIELD (CO_324:0000403) was measured in kg/ha at the plot level. All traits except FLAG were measured at harvest. Some traits, like GWGH and PAN, were not measured in all environments (Supplementary Table 5).

## Environmental covariables—ECs

We complemented the data by 15 daily observed ECs (Supplementary Table 6) to evaluate the environmental influence on plant adaptation of the GR and KK populations. With 4 environments (SB1, SB2, CZ1, CZ2) we could observe environmental gradients, which was not possible for the LT data. The ECs were divided into 3 categories. The atmospheric ECs contained: cumulated rain (mm), humidity (%), vapor pressure deficit (VPD—kPa), slope of saturation VP curve (SVP—kPa/d), potential evapotranspiration (ETP—mm/day), and atmospheric water deficit (PETP—mm/day). The temperature ECs covered: maximum and minimum temperatures (d), temperature range (d), the effect of temperature on radiation use efficiency (FRUE; 0–1), and cumulated DD. The radiation ECs were the cumulated hours of sun (hSun), the photoperiod (day length) (h), and the solar radiation (MJ/m$^2$/day). We also included the photothermal time as the product between photoperiod and DD. A principal component analysis of the environments' EC values showed important differences between the environmental conditions and a good coverage of the EC variability (Supplementary Fig. 3).

The EC values came from weather stations at the field experiments complemented by Nasapower satellite observations (Spark 2018) and transformation using the R package EnvRtype (Costa-Neto *et al.* 2021). We projected the genetic effects beyond the tested environments, using compiled environmental data from various sources extended with EnvRtype for a grid of

60 points between 12.25 and 13.75 degrees of latitude and −12.5 and −2 of longitude (Fig. 5).

## Phenotypic data analysis

We estimated the genotypic variance component and broad-sense heritability ( $h^2$) using the following mixed model:

$$y_{icjkl} = E_j + \underline{R(E)_{jk}} + \underline{B(R,\ E)_{jkl}} + C_c + \underline{G(C)_{ic}} + \underline{GE(C)_{ijc}} + \underline{e_{icjklm}} \qquad (1)$$

Where, $y_{icjkl}$ = plot phenotypic observation of the ith genotype from cross c in environment j, replication k, and block l; $E_j$ = environment effect; $R(E)_{jk}$ = replication effect within-environment (only for LT subpopulation); $B(R,\ E)_{jkl}$ = block effect within replication and environment; $C_c$ = cross effect; $G(c)_{ic}$ = genotype effect conditional on cross; $GE(C)_{ijc}$ = GxE effect conditional on cross. The underlined terms were considered as random, and the other ones as fixed. The genotype, GxE, and error terms were normally distributed with cross-specific variance ( $\sigma^2_{G(cr)}$, $\sigma^2_{GE(cr)}$, $\sigma^2_{e(cr)}$). We estimated the model components using Genstat 18 (VSN International 2022). Given those, we calculated the broad-sense $h^2$ using the formula of Hung et al. (2012):

$$h^2 = \frac{\sigma^2_{G(cr)}}{\sigma^2_{G(cr)} + \dfrac{\sigma^2_{GE(cr)}}{N_{env}} + \dfrac{\sigma^2_{e(cr)}}{(N_{env}*N_{rep})}}$$

Where, $N_{env}$ is the number of environments and $N_{rep}$ the number of replications. For the multi-environment QTL analysis, we calculated within-environment best linear unbiased estimates (BLUEs) by removing the environment and cross term from model 1 and by considering the genotype term as fixed. We used the plot field coordinates to model the spatial variation using a 2D P-spline (SpATS model, Rodriguez-Alvarez et al. 2018). We performed correlation analyses of the BLUEs to investigate the relationships between traits.

## Environmental covariable selection

For each subpopulation (GR2012, GR2013, KK2012, KK2013) by trait combination, we selected the 5 ECs that had the strongest influence on the phenotype and the time window where this effect was maximal using the method of Li et al. (2018). This method consists of calculating the correlation between the trait BLUE means across the environments and the EC values inside time windows of different sizes (20, 40, 60, 80, or 100) starting on different days of the plant cycle. We selected the 5 ECs with the highest overall average correlations and determined the most influential window and corresponding EC values as the one with the highest EC-trait correlation. Those EC values and time windows were later used in the QTLxEC models.

## MPP QTLxEC modeling

To detect QTL and characterize their effect, we performed a sequential analysis with models increasing in complexity. We started by detecting QTL positions with an extended version of the linear mixed model 3 proposed by Garin et al. (2020):

$$\underline{y_{icj}} = E_j + C_{cj} + x_{i_qp}*\beta_{pj} + \underline{GE_{icj}} + \underline{e_{icj}} \qquad (2)$$

Where, $y_{icj}$ = BLUE of genotype i from cross c in environment j. $E_j$ = environment effect and $C_{cj}$ = within-environment cross

effect. We assumed that the QTL effect was multi-allelic with a different allelic effect carried by each parent. Indeed, the potential existence of multiple alleles (Giraud et al. 2014), of multiple genomic regions controlling the trait with cross-specific differences of frequency including absence of segregation and fixation (Xu 1998; Holland 2007), as well as of difference of linkage disequilibrium in different crosses (Bauer et al. 2013) make the multi-allelic hypothesis realistic. The use of a parental model is a common procedure in MPP QTL detection (Li et al. 2011; Bardol et al. 2013; Garin et al. 2017), which presents the advantage of linking the source of the QTL effect to an identified parent (Christopher et al. 2021). Therefore, the QTL position, $x_{i_qp}$ represented the number of alleles from parent p carried by genotype i. It was inferred by looking at marker score similarity between the genotype and its parent within cross. An alternative bi-allelic model would infer genetic similarity by looking only at marker score similarity ignoring the cross information. Accordingly, $\beta_{pj}$ represents the QTL allelic effect of parent p in environment j. The recurrent parent was set as a reference. The overall QTL effect significance was estimated using the Wald test (Verbeke et al. 1997), which is a standard procedure to test for fixed effects significance in mixed models. Under H0, all parent environment-specific effects ( $\beta_{pj}$) were assumed to be equal to 0, and the test statistic followed a chi-squared distribution with N alleles degrees of freedom. Under H1, at least one $\beta_{pj} \neq 0$.

The $GE_{icj}$ term is the residual genetic by environment variation and $e_{icj}$ is the plot error term that cannot be estimated separately due to the nonreplicated nature of the BLUEs. To model the ( $GE_{icj} + e_{icj}$) term, we extended the model from Garin et al. (2020) using an unstructured variance covariance (VCOV) structure (Boer et al. 2007). The unstructured model estimates 1 (co)variance ( $\sigma^2_{Gj,\ j'}$) for each pair of environments, which requires an estimate of $N_{env}*(N_{env} + 1)/2$ parameters. For example, for 2 genotypes (i and i') measured in 4 environments the VCOV is equal to

$$V = \begin{bmatrix} y_{i.1} \\ y_{i'.1} \\ y_{i.2} \\ y_{i'.2} \\ y_{i.3} \\ y_{i'.3} \\ y_{i.4} \\ y_{i'.4} \end{bmatrix} = \begin{bmatrix} \sigma^2_{G1} & 0 & \sigma^2_{G1,2} & 0 & \sigma^2_{G1,3} & 0 & \sigma^2_{G1,4} & 0 \\ & \sigma^2_{G1} & 0 & \sigma^2_{G1,2} & 0 & \sigma^2_{G1,3} & 0 & \sigma^2_{G1,4} \\ & & \sigma^2_{G2} & 0 & \sigma^2_{G2,3} & 0 & \sigma^2_{G2,4} & 0 \\ & & & \sigma^2_{G2} & 0 & \sigma^2_{G2,3} & 0 & \sigma^2_{G2,4} \\ & & & & \sigma^2_{G3} & 0 & \sigma^2_{G3,4} & 0 \\ & & & & & \sigma^2_{G3} & 0 & \sigma^2_{G3,4} \\ & & & & & & \sigma^2_{G4} & 0 \\ & & & & & & & \sigma^2_{G4} \end{bmatrix}.$$

We detected QTL by performing a simple interval mapping to select cofactors followed by a composite interval mapping. To reduce the computational demand of the QTL detection scans, we calculated an approximate test statistic similar to the generalized least square strategy implemented by Kruijer et al. (2015; Method S1). The cofactors and final QTL were selected per chromosome using an iterative process: (1) select the most significant position; (2) exclude the position inside the window; (3) repeat until no free position is significant. The cofactor and QTL detection threshold was set at $1 - (1 - 0.05)^{\frac{1}{M_{eff}}}$, where $M_{eff}$ was the effective number of markers accounting for multiple testing determined with the procedure of Li and Ji (2005) (Supplementary Table 7: threshold values). We selected a maximum of 1 cofactor per chromosome to avoid model overfitting. The QTL exclusion window was set to 20 cM. We estimated a global R squared for the whole set of QTL as well as partial R squared for each final selected QTL position using a linear model.

After detecting QTL, we determined if the parental allelic effects composing the overall QTL effect had significant QTL by environment interactions (QEI). For that purpose, we decomposed the parental QTL effects into a main effect component ($\alpha_p$) and a QEI component ($\beta_{pj}$) that were estimated simultaneously, which give the following multi-QTL model

$$\underline{y_{icj}} = E_j + C_{cj} + \sum_{q=1}^{n_{QTL}} x_{i_qp}*(\alpha_p + \beta_{pj}) + \underline{GE_{icj}} + \underline{e_{icj}} \quad (3)$$

We tested for the significance of the $\alpha_p$ and $\beta_{pj}$ terms using the Wald test (P-value <0.05). Those tests allowed us to determine if the parental effects were significant ($\alpha_p$ or $\beta_{pj}$ significant), and if the parental alleles interacted with the environment ($\beta_{pj}$ significant). For the parents with a significant QEI (pxE), we replaced the $\beta_{pj}$ with $EC_j*S_p + l_{p\epsilon}$ to test the environmental sensitivity of the QTL allele with respect to the environmental covariate $EC_j$ (e.g. humidity) that represent the EC value in environment j associated with the sensitivity term $S_p$ and the residual effect $l_{p\epsilon}$. The fitted model was the following:

$$\underline{y_{icj}} = E_j + C_{cj} + \sum_{q=1}^{n_{QTL}} x_{i_qp}*(\alpha_p + \beta_{pj}) + x_{i_qpxE}(\alpha_p + EC_j*S_p + l_{p\epsilon}) + \underline{GE_{icj}}$$
$$+ \underline{e_{icj}} \quad (4)$$

We estimated model 4 for the 5 most influential ECs previously determined the significance of the sensitivity term $S_p$ using again the Wald test (P-value <0.05). For the parental alleles showing at least 1 significant EC interaction, we could predict the allelic effect beyond the tested environments by substituting the average $EC_e$ values over the next seasons (2014–2020) for a grid of 60 points in the QTL sensitivity equation ($Q_{eff} = \hat{\alpha}_p + EC_j*\hat{S}_p$). The estimation of model 3 and 4 effects was done using an exact restricted maximum likelihood. The methodology was added to the mppR R package (Garin *et al.* 2018).

We considered that QTL positions detected for the same trait but in different subpopulations (e.g. GR2012 and LT) represented the same "unique" QTL position if they were distant by less than 10 cM. For each trait, we iteratively identified unique QTL positions by searching for groups of QTL with a maximum distance of 10 cM between the 2 most extreme positions, starting with the positions with the smallest distance. We searched for candidate genes behind the unique QTL positions using the sorghum QTL atlas (Mace *et al.* 2019).

## Results
### Genetic diversity
Figure 2a illustrates the genetic diversity covered by the parental lines of the WCA-BCNAM population compared to a panel representative of the global sorghum diversity (Supplementary Method 2) and the parent of the sorghum US-NAM (Bouchet *et al.* 2017). Overall, the WCA-BCNAM parents covered from 69.9 to 86.8% (depending on the MAF threshold applied to the SNP dataset) of the global sorghum genetic diversity, which offers a better coverage than the parents from the sorghum US-NAM which captured 38.12–57.5% of the considered diversity (Supplementary Method 3 and Table 8). Principal component analysis of the WCA-BCNAM genetic data (Fig. 2b–d) detected 3 distinct groups corresponding to the 3 recurrent parents

(Supplementary Fig. 4). Clear subdivisions of the populations according to the donor parent race and some specific divergences from this general pattern (e.g. Hafijeka Guinea margaritiferum accession) were observed.

### Phenotypic data
The differences between the adjusted phenotypic value distributions were substantial, with medium to large standard deviations [e.g. 130 degree days (~7 days) for FLAG, 43 cm for PH, or 892 kg/ha for YIELD]. The phenotypic value distributions were also characterized by large ranges, with 47 days for FLAG and 289 cm for PH (Supplementary Fig. 5). The average heritability values over populations were larger for FLAG (0.78–0.95), NODE_L (0.8–0.9), PH (0.76–0.88), and PED_L (0.68–0.9) compared to PAN_L (0.38–0.77), NODE_N (0.4–0.66), GWGH (0.09–0.89), or YIELD (0.37–0.64) (Supplementary Table 9). Heritability values were larger in the LT subpopulation, which is due to the within-environment replication as well as the greater similarity between the environmental conditions in which the LT subpopulation was phenotyped.

In terms of correlations between the BLUEs of traits (Supplementary Figs. 6 and 7), we observed an overall negative relationship between FLAG and YIELD with an average Pearson correlation of −0.28 (P-val = 0.01) and a standard deviation of 0.17 over the different subpopulations and environments. This negative relationship was observed in all genetic backgrounds. FLAG and NODE_N were positively correlated in all genetic backgrounds. Concerning the correlation of PH with its components, the strongest 1 was with NODE_L (0.74 ± 0.12; P-val < 0.001), and the weakest with NODE_N (0.26 ± 0.14; P-val = 0.06). It took intermediary values for PED (0.53 ± 0.15; P-val = 0.06) and PAN (0.41 ± 0.13; P-val = 0.06). This pattern was observed in all configurations. PH was positively correlated with YIELD (0.23–0.56 P-val = 0–0.007), except for the GR subpopulation measured in 2013 at Sotuba (−0.11 ± 0.01 P-val = 0.13). Finally, GWGH was generally correlated with YIELD (0.27 ± 0.18 P-val = 0.03), with a stronger correlation in KK 2012 (0.46 ± 0.01 P-val < 0.001). The ECs influencing the traits the most and the time windows when the influence was maximal are listed in Supplementary Table 10 and Fig. 8.

### QTL detection—general results
The total length of the consensus genetic map was 1412 cM, with a number of cross-overs equal to 47,669, 20,343, and 20,120 in the GR, KK, and LT populations, respectively (Supplementary Table 2). We detected 100 significant QTL over the 5 subpopulations and 8 traits, which represented 64 unique QTL (Table 2, Supplementary Fig. 9). The total variance explained by the QTL was rather large for FLAG (32–53%), PH (10–48%), and NODE_L (11–47%), moderate for PED (10–32%), NODE_N (10–22%), and GWGH (8–30%), and low for YIELD (4–14%) and PAN (5–9%).

### QTLxEC extend
Each of the 100 significant QTL was modeled by assuming $N_{par(sub-pop)} − 1$ allelic effects, which represented a total of 948 effects for which we could estimate the significance of the main and GxE additive effects (Supplementary Table 11). Overall, 51% of the parental alleles presented significant effects, 26% interacted with the environment (238 alleles corresponding to 87 QTL), and 16% interacted with at least 1 EC (128 alleles corresponding to 61 QTL). The FLAG, PH, and PED QTL were more significantly affected by the EC than the QTL of PAN and YIELD (Supplementary Table 12). For example, photoperiod strongly

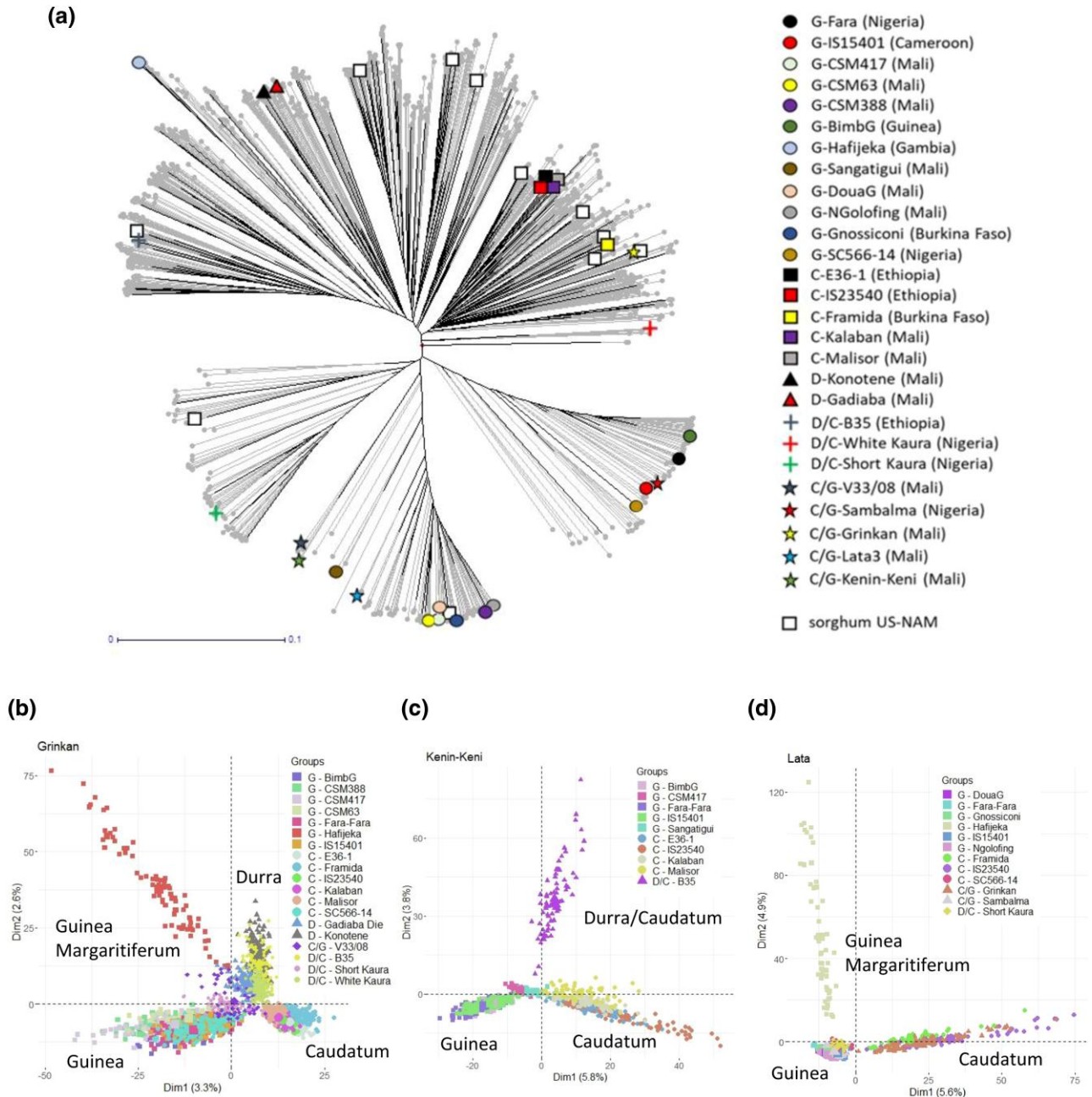

**Fig. 2.** Genetic diversity and structure of the WCA-BCNAM design. a) Coverages of the global sorghum molecular diversity by the WCA-BCNAM and sorghum US-NAM (white square, Bouchet *et al.* 2017) parents. Principal component biplots performed on a subset of 5,000 markers randomly selected from the b) GR c), KK d) LT subpopulations.

influenced FLAG, NODE_N, and PH QTL. Atmospheric ECs like VPD influenced PED and NODE_L QTL effects, while YIELD QTL was sensitive to humidity. PAN and NODE_L QTL were sensitive to minimum temperature.

## QTL with large effects and candidate genes

Eleven QTL showed medium to large effects ( $R^2 \in [2.7 - 29.9]$) with strong significance and consistency over several subpopulations and environments (Table 3 and Supplementary Fig. 9). The range of the parental allelic effects could reach 300 dd (around 15 days) for FLAG, or 1.07 m for PH, which accounts for an important fraction of the phenotypic variability (Supplementary Fig. 5). On chromosome 3, we detected a strong QTL for FLAG

(QTL_FL_3_78) which was significant in all subpopulations and environments. Almost at the same position, we also detected a large effect QTL for NODE_N (QTL_NN_3_78). QTL_FL_3_78 and QTL_NN_3_78 are probably linked to the early flowering (*Elf3*) possible candidate gene (Guitton *et al.* 2018) or *SbCN12* (Yang *et al.* 2014). Another FLAG QTL (QTL_FL_6_3) with a consistent effect in all subpopulations and environments was detected at the beginning of chromosome 6. It colocalized with a QTL for NODE_N (QTL_NN_6_2). Those QTL could be related to the *Ma6* gene (Rooney and Aydin 1999; Murphy *et al.* 2014). We also detected a QTL with medium effects on FLAG on chromosome 6 around 36 cM (QTL_FL_6_38) in the region of the *Ma1* gene (Murphy *et al.* 2011) and another on chromosome 9 around 105 cM

**Table 2.** Number of QTL detected for the traits and reference genotype by year combinations. Total $R^2$ explained by the QTL are provided in parentheses.

|  | FLAG | PH | NODE_N | NODE_L | PED | PAN | GWGH | YIELD |
|---|---|---|---|---|---|---|---|---|
| Grinkan 2012 | 6 (48.9) | 3 (48) | 4 (17.6) | 2 (47.1) | 6 (31.2) | 5 (8.5) | 3 (13.5) | 3 (5.4) |
| GR 2013 | 4 (32.1) | 4 (44.5) | 1 (9.6) | 2 (42.6) | 2 (21.1) | 1 (4.6) | 0 | 2 (7.3) |
| KK 2012 | 6 (53.4) | 2 (9.6) | 2 (16.8) | 3 (18.8) | 3 (14.3) | 2 (5.2) | 2 (7.9) | 1 (3.9) |
| KK 2013 | 2 (35.5) | 2 (12.7) | 2 (22.3) | 0 | 2 (16.9) | 1 (5.8) | 0 | 1 (5.9) |
| LT | 4 (50.3) | 2 (20.1) | 2 (13.7) | 1 (11) | 2 (10.2) | 0 | 6 (30.4) | 4 (14.3) |

**Table 3.** List of large and medium effect QTL with trait, chromosome, position, average $R^2$, QTLxE effect range, number of parental alleles with significant effects, and candidate genes.

| QTL ID | Trait | chr | Range (cM) | Range (Mbp) | $R^2$ | QxE range | Npar | Candidate genes |
|---|---|---|---|---|---|---|---|---|
| Q_FL_3_78 | FLAG | 3 | 77.34–78.36 | 51.1–51.5 | 17.1 | (−123; 144) (dd) | 21 | *Elf3, SbCN12* |
| Q_NN_3_78 | NODE_N | 3 | 78.13–78.75 | 51.4–51.7 | 9.3 | (−1.8; 2.1) (n) | 15 | *Elf3, SbCN12* |
| Q_FL_6_3 | FLAG | 6 | 1.49–2.94 | 0.4–0.8 | 19.4 | (−178; 130) (dd) | 17 | *Ma6* |
| Q_NN_6_2 | NODE_N | 6 | 1.49–2.73 | 0.4–0.8 | 7.9 | (−2.6; 1.8) (n) | 15 | *Ma6* |
| Q_FL_6_38 | FLAG | 6 | 36.32–39.4 | 40.4–41.2 | 6.3 | (−192; −27) (dd) | 6 | *Ma1* |
| Q_FL_9_105 | FLAG | 9 | 103.7–106.7 | 54.6–55.4 | 2.7 | (−40; 91) (dd) | 9 | *SbFL9.1* |
| Q_PH_7_76 | PH | 7 | 74.28–76.69 | 54.7–55.2 | 21.9 | (−37; 69) (cm) | 16 | *qHT7.1, (Dw3)* |
| Q_NL_7_78 | NODE_L | 7 | 76.29–79.59 | 55.1–55.8 | 29.9 | (−0.1; 4.2)) (cm) | 14 | *qHT7.1, (Dw3)* |
| Q_PED_7_78 | PED | 7 | 74.8–82.1 | 54.84–56.26 | 12.6 | (−8; 10) (cm) | 21 | *qHT7.1, (Dw3)* |
| Q_NL_7_98 | NODE_L | 7 | 96.1–100.7 | 58.3–59.1 | 11.8 | (−5.8–4.2) (cm) | 9 | *Dw3* |
| Q_PH_7_106 | PH | 7 | 102–108.3 | 59.4–60.7 | 7.6 | (−14; 58) (cm) | 15 | *Dw3* |

(QTL_FL_9_105) potentially close to the *SbFL9.1* gene (Higgins *et al.* 2014; Bouchet *et al.* 2017).

A strong QTL for PH has detected on chromosome 7 around 75 cM (QTL_PH_7_76) with significance in the GR2013 and KK2013 subpopulations. This QTL colocalized with a highly significant QTL for NODE_L (QTL_NL_7_78) and a strong and highly consistent QTL for PED (QTL_PED_7_78). Nearby this QTL, another QTL (QTL_PH_7_106) also had a large effect on PH and colocalized with a large effect QTL for NODE_L (QTL_NL_7_98). The QTL region of chromosome 7 could be related to 1 or 2 genes. The main candidate gene is *Dw3* (Multani *et al.* 2003; Brown *et al.* 2008). However, according to Li *et al.* (2015), chromosome 7 could harbor 2 genes: *qHT7.1,* positioned before *Dw3*, may influence both stem length and peduncle length, while *Dw3* may only influence stem length.

## Complex QTL effect pattern at large effect QTL

QTL with large effects showed complex patterns with effects distributed over many parents and a wide range of effects modulated by the genetic background and the environment. Almost all parental alleles had at least 1 significant EC interaction. Such a complex pattern can be illustrated for QTL_FL_3_78 (Fig. 3). First, we noticed the contrasting parental effects with parents like CSM417 or CSM388 whose alleles reduced maturity, while IS15401 alleles increased it. Then, we observed differences in expression due to the genetic backgrounds. For example, the allele of Fara-Fara had a small effect in a GR background, while it strongly increased maturity in the LT and KK backgrounds. Finally, we also observed environmental differences, like the stronger cycle reduction of the CSM388 allele in 2012 compared to 2013.

## QTL effect on photoperiodism

The plots of Fig. 4a–c represents QTL allele additive effects for a given photoperiod compared to the recurrent (reference) parent. QTL_FL_6_3 (*Ma6*) was the QTL with the highest proportion of alleles influenced by the photoperiod (Fig. 4a). At that position

compared to GR, the effects of E36-1 or B35 were negatively affected by the photoperiod. Those alleles reduced the photoperiod sensitivity. When the days were longer (e.g. at the first sowing date ~12.5 h) a genotype carrying those alleles increased its cycle less than a genotype carrying the allele of the central parent. QTL_FL_6_38 (*Ma1*) was also sensitive to photoperiod with 5 parental alleles interacting significantly with the photoperiod over the different genetic backgrounds. For example, in GR2012, the allele of White Kaura increased the photoperiod sensitivity compared to the recurrent parent (Fig. 4b).

At QTL_FL_3_78 (*Elf3*, Fig. 4c), the parental alleles were mostly insensitive to photoperiod. We only detected significant effects of the photoperiod for 3 alleles. The alleles of Fara-Fara and IS15401 had a reduced sensitivity, while the allele of B35 increased it. The alleles of QTL_FL_9_105 (*SbFL9.1*) were also insensitive to the photoperiod with only 2 significant interactions. Even though the EC range looks small (0.3 h), it caused important variations of the QTL effect. For example, at QTL_FL_6_36, the effect of an SC566-14 allele varied by 60 dd (around 3 days; Fig. 4b).

## Dissecting PH genetic determinism

The phenotypic data for PH and its components (NODE_N, NODE_L, PED, PAN) allowed us to dissect PH genetic architecture. PH can be expressed as PH = (NODE_N * NODE_L) + PED + PAN. Since the phenotypic values of NODE_N are strongly correlated with FLAG, it was not surprising to find overlapping QTL for the 2 traits on chromosomes 3 and 6 (QTL_NN_3_78, QTL_NN_6_2). In terms of photoperiod sensitivity, the QTL for NODE_N followed a similar pattern to the ones from FLAG. QTL_NN_3_78 (*Elf3*) was rather insensitive to photoperiod with 4 parental alleles having a significant interaction, while QTL_NN_6_2 (*Ma6*) was more sensitive with 6 alleles with significant interactions with the photoperiod (e.g. Malisor 84-7).

We also observed a strong agreement between the QTL positions detected for PH and NODE_L on chromosome 7. QTL_PH_7_76 and QTL_NL_7_78 (*qHT7.1*) colocalized, while

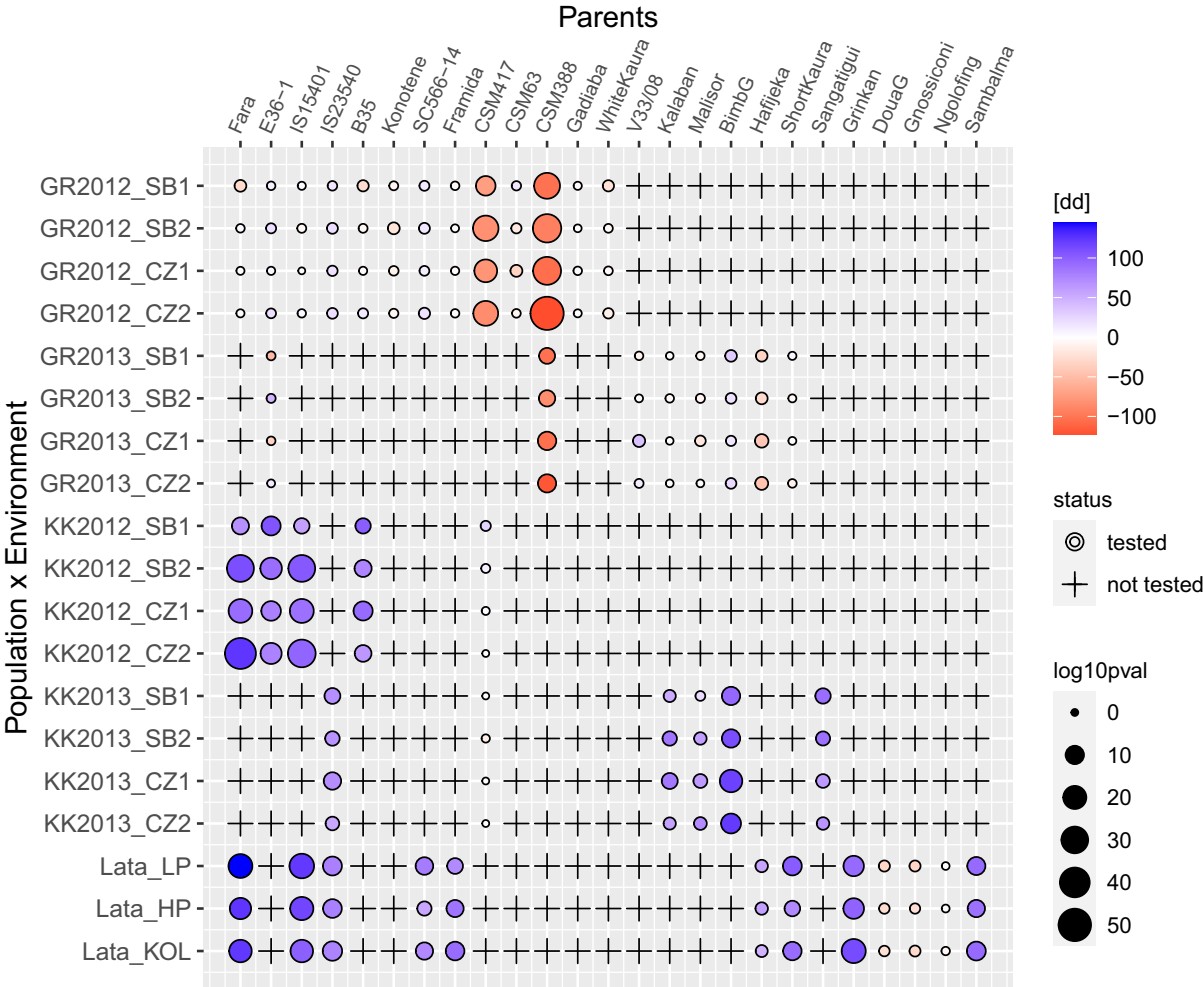

**Fig. 3.** Estimated QTL allelic effect given environment and genetic background for flag leaf appearance using model 2 at chromosome 3 position 77.3 cM. X axis: donor parents. Y axis: genetic backgrounds (GR, KK, LT) and environments (SB1-2: Sotuba sowing 1 and 2, CZ1-2: Cinzana sowing 1 and 2, LP/HP: low/high phosphorus, KOL: Kolombada). The color is proportional to the effect size and the circle size to its significance. Crosses represent untested allele✕background✕environment combinations.

QTL_PH_7_106 and QTL_NL_7_98 (*Dw3*) were separated by less than 10 cM. The QTL influencing NODE_L were not sensitive to photoperiod, but other ECs like VPD or potential ETP modulated the parental allelic effects at those positions. For example, at QTL_PH_7_76, the effects of parents like Hafijeka or Short Kaura were reduced when VPD increased (Fig. 4e). Surprisingly, the corresponding QTL (QTL_PH_7_76) detected for PH showed significant interaction with the photoperiod (Fig. 4d) in the GR subpopulations. We consider that this apparent effect of photoperiod on QTL_PH_7_76, is due to the fact that PH is proportional to the interaction NODE_N * NODE_L. Therefore, at chromosome 7, the signal is due to the interaction between a photoperiod-sensitive component (NODE_N) and a photoperiod-insensitive part (NODE_L). We hypothesize that the apparent photoperiod sensitivity of QTL_PH_7_76 is due to the NODE_N component of the height that is strongly photoperiod-sensitive, for example via the QTL on chromosome 6 (QTL_NN_6_2).

For PED, QTL_PED_7_78 (*qHT7.1*) was the most environmentally sensitive QTL. This QTL was not photoperiod-sensitive, but covariables like SVP had a negative effect on the propensity to increase PED compared to the reference parent. This effect was consistent in the GR2013 subpopulation with 5 parents (V33/08, BimbG,

Kalaban, Hafijeka, and Short Kaura) reducing their propensity to increase PED when SVP increased (Fig. 4f). It is interesting to emphasize that drought-related ECs (VPD, SVP) influenced both QTL_PH_7_76 and QTL_PED_7_78 with effects going in the same direction.

## Expected QTL effect beyond the tested environments

Between 12 and 13.75 degrees of latitude, the Malian environment is characterized by a southwest-to-northeast gradient (Supplementary Fig. 2). The southwest is cooler with lower temperature ranges, higher precipitation, and humidity, while the northeast is drier with higher temperature ranges and lower precipitations. A final extension of our results is the projection of QTL allelic effects having a significant interaction with 1 of the ECs in the Malian environment. For that we substituted the observed EC values from a grid of 60 points in the estimated allele sensitivity equation (Fig. 5). Those results represent the expected QTL allele additive effect given the new EC conditions with respect to the central parent in the environment with the lowest EC value.

In Fig. 4d and f, we represented the expected behavior of the BimbG and V33/08 alleles at QTL_PH_7_76 and QTL_PED_7_78,

**Fig. 4.** Sensitivity of the QTL parental allele effects to environmental covariables calculated with model (4). Each slope is described by $\hat{a}_p + EC_j * \hat{S}_p$, with $\hat{a}_p$ representing the main allelic additive effect and $\hat{S}_p$ the sensitivity to increase in the environmental covariate ($EC_j$). a) Flag leaf QTL chr 6 (3 cM) in GR2012 given photoperiod; b) Flag leaf QTL chr 6 (36.3 cM) in GR2012 given photoperiod; c) Flag leaf QTL chr 3 (77.3 cM) in KK2012 given photoperiod. d) PH QTL chr 7 (76 cM) in GR2013 given photoperiod. e) PH QTL chr 7 (76 cM) in GR2013 given temperature range. f) Peduncle length QTL chr 7 (78 cM) in GR2013 given SVP. g) Illustration of the relationship between flowering time, PH, and photoperiod with influential genomic regions on the right side. Flowering time can be seen as a broken linear function with a constant baseline period and a photoperiod-sensitive period when time to flowering increases with photoperiod. Plants with early (late) flowering time due to an extended baseline period (Elf3) and/or an increased photoperiod sensitivity (Ma6, Ma1) become shorter (taller) due to the accumulation of fewer (more) internodes (Elf3, Ma6) whose average size (red arrow, qHT7.1, Dw3) is not influenced by the photoperiod. The peduncle length (green arrow) is also not photoperiod-sensitive (qHT7.1).

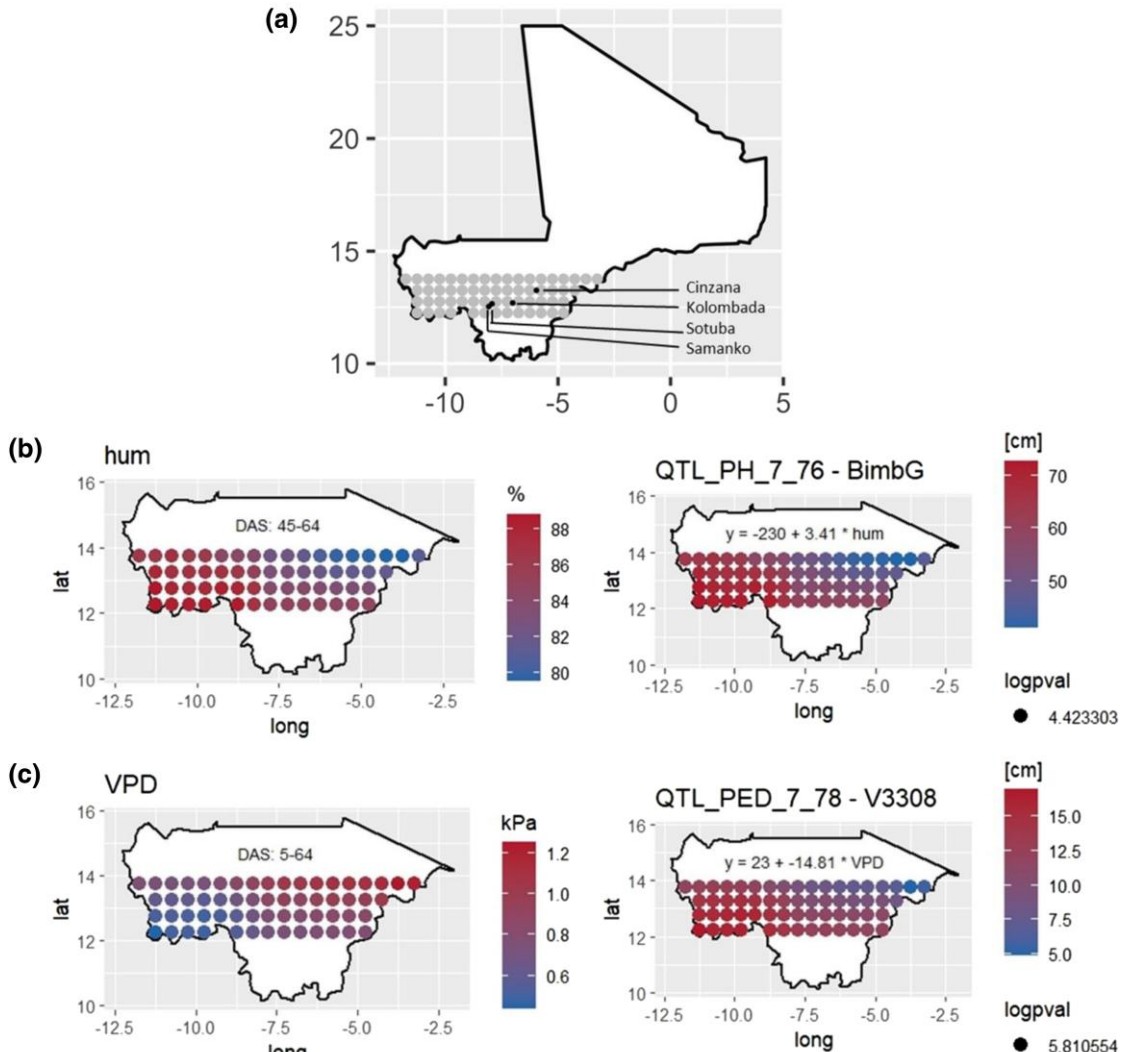

**Fig. 5.** Extrapolating the QTL by EC at the Malian scale a) map of Mali with testing locations and neighboring areas of projection. b and c) Projections of additive QTL allelic effects given allelic sensitivity and new VPD and humidity conditions. The effect must be interpreted as the deviation with respect to the central parent in the environment with the lowest EC value. QTL positions: b) QTL_PH_7_76 BimbG×humidity; and c) QTL_PED_7_78 V33/08×VPD. The projections were obtained by substituting observed environmental covariables in a grid of 60 points of the Malian environment in QTL allele sensitivity equations.

respectively. The BimbG allele was positively influenced by humidity, which increases its effect on PH in the more humid southwest and reduces it in the drier northeast regions (Fig. 5b). Figure 5c illustrates the effect of VPD on the effect of the V33/08 allele at QTL_PED_7_78 on PED extension. PED extension was reduced in the northeast drier regions, while it was increased in the more humid southwest part of Mali. We can emphasize that those 2 alleles react similarly to the environmental gradient by increasing PH more in the southwest.

## Discussion

The main objectives of this work were the introduction of a new sorghum genetic resource and a new statistical methodology to dissect QTLxE interactions. Here, we emphasize the properties of the Western and Central Africa BCNAM design and of the proposed statistical methodology, which constitute a lever to support the development of improved varieties. Although grain yield and GWGH have been analyzed, only a limited set of QTL and QTLxEC interactions have been identified for those traits. This

likely results from a lack of power due to a too small set of experiments combined with moderate broad-sense heritability of these traits. In this context, the final sections of the discussion are focused on flowering time and PH, 2 traits well recognized as key components of the adaptation to local abiotic conditions (Buckler *et al.* 2009; Moles *et al.* 2009).

### Multireference BCNAM design properties

A major contribution of this work is the development of a new sorghum genetic resource taking the form of a multireference BCNAM design. Its usefulness can be evaluated given criteria like genetic diversity, mapping power and resolution, and potential for genetic gains. In terms of diversity exploration, the use of 3 recurrent parents instead of one, as in almost all the (BC)NAM populations, substantially increases the exploitable genetic diversity. This extra diversity increases the contrast between the recurrent and the donor parent, which should help to detect significant genetic regions and estimate their environmental sensitivity.

In terms of QTL detection, the usefulness of a design can be evaluated in terms of detection power, capacity to estimate and

trace the QTL effect, and mapping resolution. The power gain offered by the NAM design compared to biparental populations (Li *et al.* 2011) or association panels (Bouchet *et al.* 2017) has already been demonstrated. This should also be valid for our population. The most relevant advantage of a multireference BCNAM design is the possibility to test allelic effects in several genetic backgrounds. Similarly to Christopher *et al.* (2021), we show that the parent allelic effect can be strongly modulated by the genetic background (Fig. 3). More generally, in terms of QTL effect characterization, MPPs like the (BC)NAM designs increase the allele frequencies and allow the user to trace back the allelic effect to a specific parent (Myles *et al.* 2009). Analyzed properly, such data allow the precise estimation of the parental allelic effect, which is fundamental for breeding applications.

The main disadvantage of the (BC)NAM design is the low mapping resolution compared to designs involving further intercrossing like MAGIC (Klasen *et al.* 2012). Even though the (BC)NAM design involves more recombination than biparental populations, the recombinations are still restricted within a cross, which considerably extends the linkage disequilibrium decay (Garin *et al.* 2021). The combination of our design with strategies like RapMap (Zhang *et al.* 2021) could improve the resolution. In terms of genetic gain, Bernardo (2021) showed that populations like MAGIC do not have a significant advantage compared to multiple cross populations like the BCNAM. Designs like the (BC)NAM population can also increase the prediction ability of untested lines in genomic selection schemes due to the genetic relatedness between lines composing the population (Lehermeier *et al.* 2014).

A last question related to the multireference BCNAM design optimization is the need to cross all donor parents to the recurrent parents (full factorial) or only a subset. On the one hand, by performing all crosses we improve our ability to estimate QTL allelic effects in multiple genetic backgrounds and determine the stability of those effects. On the other hand, given a fixed total population size, the use of a partial factorial design could increase the individual cross sizes, which increases the QTL detection power (Garin *et al.* 2021). Reducing the number of crosses per parent could also increase the number of parents included in the design and therefore the addressed genetic diversity. Given that the estimation of the QTL effect is conditioned by its detection, we consider that performing only a selected number of crosses in a multireference (BC)NAM design could be an interesting strategy to increase the QTL detection power and/or diversity. More definitive answers regarding the optimization of multireference BCNAM designs could be obtained by simulations.

## Statistical methodology properties and limitations

In this work, we propose an approach allowing the integration of high-resolution environmental information in MPP-ME QTL analyses to improve our understanding of the GxE interaction. Few alternative approaches analyze this kind of data (e.g. De Walsche *et al.* 2023). We propose a sequential strategy to identify parental alleles exhibiting GxE interactions and decompose them into environmental covariate sensitivity curves. We used a mixed model with unstructured VCOV structure to control for the genetic (co)variance and performed an approximate model estimation for the QTL scan. However, we performed an exact model estimation of the QTL effects and their interaction with the environment. Such a joint effect estimation could be more precise than the meta-analysis of a single environment proposed by De Walsche *et al.* (2023).

The main limitation to illustrate our approach was the number of available environments, especially for the QTLxEC sensitivity

estimation. Even if the selected environments cover a broad variability, their reduced number limits the statistical inference of our models. From a general point of view, 2 environments are sufficient to estimate QTL effect variations. However, our extension of the QTLxE effect to estimate its sensitivity to the environmental covariates requires that each genotype be observed in a number of conditions that are large enough to estimate the sensitivity curve parameters. Statistically, the available degrees of freedom to estimate the $\hat{\alpha}_p$ and $\hat{S}_p$ terms are equal to $N_{env} - 2$. This forced us to test for a maximum of 1 EC at a time, and to interpret our results with caution. Our strategy still allowed us to establish whether the QTL environmental variation could be explained by some of the ECs, but it was often difficult to make precise distinctions between several ECs showing similar degrees of correlation with the QTL effect variation.

The combination of genetic analysis and crop modeling is an alternative and complementary approach. The construction of a genotype-to-phenotype (G2P) set-up integrating QTL expression of crop model parameters potentially allows the extrapolation of the QTL effect to any environment (Reymond *et al.* 2003). Such a G2P set-up can support the design of new ideotypes (Gu *et al.* 2014), and the prediction of new genotype behavior in future climate scenarios (Kadam *et al.* 2019). This strategy builds on the knowledge accumulated in the crop model to reconstruct the GxE interaction, while our QTLxEC approach is based on less complex physiological relationships derived from observed data that make the prediction dependent on the observed conditions (Bustos-Korts *et al.* 2016). However, the success of the G2P approach is based on the identification of less complex physiological mechanisms that are stable across environments (Tardieu 2003). The crop model parameters are often difficult to phenotype and require controlled environments that strongly differ from natural conditions (Gu *et al.* 2014).

## Variations in flowering time genetic architecture and its GxE component

Our method helped us to gain insight into the genetic architecture of flowering time in the emerging C4 model sorghum. Our results showed that, in this experimental design, flowering time was controlled by 3–4 QTL with very strong effects. Those effects were consistent in most of the genetic backgrounds and environments but could be strongly modulated by ECs, especially the photoperiod. Those observations strongly contrast with those of Mace *et al.* (2013), who observed that flowering time was controlled by many small loci with a low proportion of GxE due to the temperature rather than the photoperiod.

Despite important similarities with Mace *et al.* (2013), like the use of a similar BCNAM design covering a large genetic diversity and the characterization in a large spectrum of environments producing important phenotypic differences, the selection applied during the population development and the statistical methodology can explain observed differences. Compared to us, Mace *et al.* (2013) applied a stronger control of the flowering time range during the development of their population. They considered that this selection strongly reduced the effect of important genes involved in the photoperiod pathway like Ma1, which can explain why most of the detected genomic regions were involved in the temperature pathway and the reduction of the observed GxE interaction. Our moderate control of flowering time certainly allowed the preservation of strong allelic effects involved in the photoperiod pathway.

Next to the differences in selection pressure, we also want to emphasize the difference in terms of statistical approach used.

We estimated models using within-environment adjusted means, while Mace *et al.* (2013) used across-environment adjusted means which do not allow the estimation of QTLxE effects (Garin *et al.* 2020). Even though we estimated a strong heritability for flowering time, which as in Mace *et al.* (2013) indicates a low level of GxE, our QTLxEC model still allowed us to detect important modulation of QTL effects by ECs like the photoperiod.

Using our QTLxEC models, we could further characterize the genetic basis of flowering time. For example, we could illustrate the photoperiod-insensitive nature of QTL_3_FL_78 (*Elf3*; same result as Guitton *et al.* 2018) and QTL_FL_9_105 (*SbFL9.1*) which only influenced the baseline duration. On the other hand, the QTL on chromosome 6 (QTL_FL_6_3 and QTL_FL_6_38) linked to the Ma6 and Ma1 regions influenced the baseline flowering duration, but also had effects influenced by the photoperiod, which makes them a source of photoperiod sensitivity (Fig. 4g). QTL_FL_6_3 (Ma6) was mostly influenced by the radiation covariables (photoperiod, observed hours of sun, and solar radiation), which support the hypothesis of Ma6 being part of the photoperiod pathway (Takai *et al.* 2012). Concerning the Ma6 region, we should also emphasize that, according to the original hypothesis of Rooney (1999), it is supposed to influence flowering time in an epistatic way with the Ma5 gene, located on chromosome 2 (Mace and Jordan 2010). We detected a strong signal in the region of Ma6, but only a mild signal on chromosome 2 (Ma5 region) in the GR subpopulation tested in 2012. This absence of signal on chromosome 2 could be due to a fixation of the Ma5 gene to the favorable allele in our subpopulations, or to the fact that in our population Ma6 mostly affects flowering time independently of other genomic regions.

Finally, we observed that QTL_FL_6_38 (Ma1) was significantly influenced by both photoperiod-related ECs and the temperature range. In our trials, the temperature range was strongly influenced by the sowing date (i.e. the photoperiod), therefore it is difficult to disentangle their respective effects. We still hypothesize that QTL_FL_6_38 (Ma1) is part of the photoperiod pathway (Childs *et al.* 1997; Rooney 1999; Higgins *et al.* 2014), but it is possible that temperature also influences this genomic region.

### PH components and their interactions with the environment

Our genetic resources and methodology helped us to gain knowledge about genetic regions strongly influencing PH. We showed that, in the considered design, PH was almost exclusively influenced by 2 QTL on chromosome 7 (QTL_PH_7_76 and QTL_PH_7_106). Those QTL cover the regions of the previously identified candidate genes *qHT7.1* and *Dw3* (Li *et al.* 2015; Bouchet *et al.* 2017). The decomposition of PH into the number and length of internodes, peduncle length, and panicle length showed that PH is controlled by phenology-dependent mechanisms influencing the number of internodes and phenology-independent genes controlling the length of the internode. The colocalization of QTL for the number of internodes and flag leaf appearance on chromosome 3 (QTL_FL_3_78 and QTL_NN_3_78) and 6 (QTL_FL_6_3 and QTL_NN_6_3) illustrates the connection of PH and plant cycle via the potential pleiotropic action of the Elf3 candidate gene region Ma6, and to a lesser extent Ma1. The genetic association between flowering time and the number of internodes makes sense because internode organogenesis is a function of the plant cycle (Takai *et al.* 2012; Fig. 4g). Given sufficient nutrients, longer maturity allows the plant to accumulate more internodes.

Other important genetic determinants of PH were located on chromosome 7 with very strong effects on the length of the internode (QTL_NL_7_78 and QTL_NL_7_98) and on the peduncle

length (QTL_PED_7_78). Our data support the results of Li *et al.* (2015) concerning the existence of 2 distinct genes (*qHT7.1* and *Dw3*) because the QTL effects of chromosome 7 were detected at different positions (around 75 cM and 100 cM) in different populations. The phenotypic effects of those positions were also consistent with the observations of Li *et al.* (2015) because the 75 cM position influenced both internode length and peduncle length, while the 100 cM position only influenced internode length. The genomic region controlling the length of the internode and the peduncle length interacted significantly with a large number of ECs. Contrary to the flowering time QTL, the QTLxEC pattern was more heterogeneous with parental alleles interacting with drought-related ECs like VPD, SVP, and humidity- or temperature-related ECs like minimum temperature or cumulated degree days. The sensitivity of sorghum peduncle length to humidity was already detected by Klein *et al.* (2001).

### Developing climate-smart varieties

The precise characterization of the genetic architecture of flowering time and PH for given environmental constraints could support the development of climate-smart varieties. We showed that adaptation mechanisms are controlled by large effects of QTL related to identified genes and candidate genes, but, like other studies, our analyses revealed that the effects of those QTL positions were distributed over many parental alleles with differences of expression given the genetic background (Mace *et al.* 2013; Higgins *et al.* 2014; Christopher *et al.* 2021). The possibility of estimating a wide range of parental effects in the WCA-BCNAM population enables the use of allelic effects for different breeding applications. For example, changing the duration of the cycle without affecting the photoperiod sensitivity (e.g. QTL_FL_3_78 CSM417: −80 dd) or varying PH via the internode length independently of the environment (e.g. QTL_NL_7_78 IS23540: +3.1 cm). All results including QTL allelic effect sensitivity were compiled in an R application (sorghum QTL effects - SQE) to facilitate their use by geneticists and sorghum breeders (https://github.com/vincentgarin/SQE).

## Conclusion

We present here a large multireference BCNAM design with unprecedented coverage of the sorghum genetic diversity of WCA. We developed a multiparent multi-environment QTL methodology that was applied to analyses of flowering time and PH that allowed us to confirm the existence of genomic regions with strong effects, like Ma1, Ma6, or Dw3. The integration of ECs in the QTLxE detection model allowed the identification of genomic regions covering Ma6 and Ma1 that are photoperiod-sensitive, while the region on chromosome 3 covering the possible candidate gene Elf3 is photoperiod-insensitive. We also reached a better understanding of PH construction through the identification of photoperiod-insensitive regions impacting the number of internodes (Elf3) and the internode size (dw3, qHT7.1) and photoperiod-sensitive regions contributing to the number of internodes (Ma1).

## Data availability

The genotypic and phenotypic data are available here https://doi.org/10.18167/DVN1/TZVGLS. The scripts to reproduce the results are available here https://gitlab.cirad.fr/agap/giv/sorghum_bcnam_analysis. The results were gathered into an interactive database available here https://github.com/vincentgarin/SQE.

For plant material, please contact the corresponding authors for availability.

Supplemental material available at GENETICS online.

## Acknowledgments

The authors thank Madina Diancoumba and Jan Jarolímek for providing the environmental data that were used to perform the projection in the Malian environment. We also thank Jean-Christophe Glaszmann and the AdaptGrass project for the access to the sorghum whole-genome sequence data.

## Funding

This work was supported by a grant from the Generation Challenge Programme (Project Numbers G7010.05.01 and G7010.05.02). The work of Vincent Garin was supported by a grant from the Swiss National Science Foundation (Postdoc.Mobility grant no: P500PB_203030). Dr Kholovà's contribution was financed by an internal grant agency of the Faculty of Economics and Management at the Czech University of Life Sciences Prague (Grant Life Sciences 4.0 Plus no. 2022B0006)

## Conflicts of interest

The authors declare no conflicts of interest.

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

*Editor: H. Daetwyler*