## [Peer Review File · Genetics]

Characterization of adaptation mechanisms in sorghum using a multi-reference back-cross nested association mapping design and envirotyping

Vincent Garin, Chiaka Diallo, Mohamed Tekete, Korotimi Thera, Baptiste Guitton, Karim Dagno, Abdoulaye Diallo, Mamoutou Kouressy, Willmar Leiser, Fred Rattunde, Ibrahima Sissoko, Aboubacar Toure, Baloua Nebie, Moussa Samake, Jana Kholova, Julien Frouin, David Pot, Michel Vaksman, Eva Weltzien, Niaba Teme, and Jean-François Rami

NOTE: The reviews and decision letters are unedited and appear as submitted by the reviewers. In extremely rare instances and as determined by a Senior Editor or the EIC, portions of a review may be redacted. If a review is signed, the reviewer has agreed to no longer remain anonymous. The review history appears in chronological order.

Review Timeline:

Submission Date:	2023-7-10
Editorial Decision:	2023-9-05
Resubmission Received:	2023-10-20
Editorial Decision:	2023-11-18
Resubmission Received:	2023-12-15
Accepted:	2023-12-20

September 5, 2023

GENETICS-2023-306315

Characterization of adaptation mechanisms in sorghum using a multi-reference back-cross nested association mapping design and envirotyping

Dear Dr. Rami:

Two experts in the field and an early career reviewer (R1) have reviewed your manuscript, and I have read it as well. The manuscript present a novel way to detect QTL by EC interactions. While your manuscript is not currently acceptable for publication in GENETICS, we would welcome a substantially revised manuscript. Both reviewers have comments and concerns to be addressed in a revised manuscript. You can read their reviews at the end of this email.

The main revisions requested are to increase clarity of writing (e.g. be very clear about the different QTL models that were fitted) and edit the manuscript to shorten some sections. The discussion seems especially long and could be more concise. We look forward to receiving your revised manuscript. Please let the editorial office know approximately how long you expect to need for revisions.

Upon resubmission, please include:

1. A clean version of your manuscript;
2. A marked version of your manuscript in which you highlight significant revisions carried out in response to the major points raised by the editor/reviewers (track changes is acceptable if preferred);
3. A detailed response to the editor's/reviewers' feedback and to the concerns listed above. Please reference line numbers in this response to aid the editor and reviewers.

Your paper will likely be sent back out for review.

Additionally, please ensure that your resubmission is formatted for GENETICS
<https://academic.oup.com/genetics/pages/general-instructions>

Follow this link to submit the revised manuscript: <https://genetics.msubmit.net/cgi-bin/main.plex?el=A2NR6FeU7A4Eyt5I6A9ftdsaOB57nesb3fzx5YVY2rgZ>

Sincerely,

Hans Daetwyler
Associate Editor
GENETICS

Approved by:
Mario Calus
Senior Editor
GENETICS

Reviewer #1 (Comments for the Authors (Required)):

The manuscript "Characterization of adaptation mechanisms in sorghum using a multi-reference back-cross nested association mapping design and envirotyping" presented a multi-reference sorghum back-cross nested association mapping populations composed of 24 diverse parents anchored on three West African elite lines. The sub-populations were tested for the QTLs of 8 phenotypic traits as well as the interaction between the phenotype and environmental covariables. The experiments are well-designed and performed, and the conclusions are appropriate to the results presented. I don't have any main concerns about the manuscript.

Reviewer #2 (Comments for the Authors (Required)):

The paper describes a new method to detect QTLxEC interaction in BCNAM populations. In general, the method looks convincing but my main concern is the small number of environments used to demonstrate the new method. In general, the paper needs significant English correction (especially the introduction). It is hard to follow in many locations. Moreover, the length of the paper can be significantly reduced. The introduction, results and discussions all has lots of unnecessary paragraphs/statements that can be removed. It looks like that paper was copied from a thesis.

Abstract

Line 58-59: "It requires advanced crossing schemes, multi-environment characterization and relevant statistical tools" Please remove this statement as it is not fully accurate. Your approach is one excellent way to do it but it can also be done with diverse populations with or without phenotypic data (e.g. detecting selective sweeps)

Line 62: "The population was characterized in environments contrasting for " Please rewrite, not very clear

Line 67: "We mobilized ..." do you mean Utilised?

Introduction

The introduction doesn't need to have subheadings.

83-84: "and therefore genetic markers" remove

Materials and Methods

It is not clear if the populations were analysed independently or jointly

The number of environments is small to have a good inference

Results

Line 379: Not all traits seem to have heritability estimation

Line 403: "47'669, 20'343, and 20'120" I think you mean "47,669, 20,343, and 20,120"

Line 404: "100 significant QTL over the five populations for eight traits, which represented 64 unique QTL" What make them unique? and are they 5 or 3 populations?

Line 405: "Consistently with the heritability estimates" this is not true, e.g. PAN. This has no relation to the heritability, it is more about the sum of effects of the major genes that can pass the significant threshold

Line 410: "The 100 significant QTL covered 1056 parental alleles" are you considering all significant SNPs? Please provide more details why you have >1000 alleles

Line 412: "and around 25% interacted with the environment" how many QTL? the same question applies for the next sentence.

The presentation of the QTL and interaction results is very long and boring. The authors could benefit from reducing this section.

Discussion

The paper would benefit from adding a discussion about the effect of the small number of environments used in the present study

Line 633: "pulblished" should be published

660-661: It is not clear how strong the effects are. They are representing high proportion of the phenotypic variation but the phenotypic variation could be small, which would let minor genes with small absolute effect values appear as major genes. Please investigate more on this, it would be a good idea to add a supplementary graphs showing the distribution of the phenotypic data.

The authors may benefit from adding a small conclusion paragraph

Supplementary method S1:

Please change "computational power" to "computational demand"

Reviewer #3 (Comments for the Authors (Required)):

The authors present a nested association mapping population of sorghum, consisting of 3,901

(BC1F4) lines derived from 24 parents crossed to 3 recurrent parents. The population was genotyped by sequencing, providing 51,545 markers, and 8 traits were phenotyped in 4 locations across 2 years with 2 sowing time treatments and 2 phosphorus levels. The locations/treatments/crosses/years are often non-overlapping (e.g., phosphorus treatments at one location in one year for crosses with one recurrent parent) but the two sowing time treatments were performed at two locations in two years for crosses with two recurrent parents (crosses and years are semi-overlapping), which forms the core of the analysis. The authors detect QTLs and focus on interactions between QTLs and environmental covariates such as photoperiod. With a model of the sensitivity of QTLs to environmental covariates, the QTL effect can be projected beyond the measured environments. The size of the population, genetic data, and phenotyping data are impressive strengths of the study.

The interactions between QTL effects and environmental covariates shown in Figure 4 (differences in line slopes) look quite subtle. If I understand correctly (I would recommend making this part of the methods clearer and more explicit), there were three QTL models: (1) QTL model where β_{pj} in equation 2 doesn't vary across environments j (2) QTLx E model where β_{pj} varies across environments j (3) QTLx EC where β_{pj} becomes a function of sensitivity to environmental covariates (ECs) as described on L344. The last model (3) was fit for 5 ECs if the p value was lower for model (2) versus model (1) (L342). I'm not clear on the method used to determine if the EC interactions were significant (L348). In general, I don't think an explicit method was used for model comparison (e.g., LR test) beyond the comparison of p values, which makes it hard to interpret the (statistical) significance of the subtle changes in slope shown in Figure 4. The effect size also suggests caution about the agronomic significance of environmental sensitivity relative to the main effect of the QTLs.

Specific comments:

Figure 3: the use of a dot and a circle is a little confusing because some small circles can look like dots. Perhaps a cross could be used instead of a dot. The parent names are also difficult to read, I suggest adjusting them so that they begin at the tick mark.

Figure 4A-F: I think the only parental effects shown are those that have lower p values under the QTLx E model. It would be nice to also display the other parental effects (i.e., those that have no significant effect relative to the recurrent parent and those that don't show evidence of QTLx E). I think the model of environmental sensitivity comes from four environments (two locations and two sowing dates). Ideally, we could also see the trait values in these four environments or, at least, note the EC value at these locations/sowing dates on the X axis. I'm also slightly confused about the EC range because Table S4 suggests the maximum range for measured photoperiod should be 11.9-12.3.

Figure 4G: This is very difficult to understand without further explanation.

Figure 5: Since these are split by QTL effects, isn't the quantity of interest the deviation from the reference genotype, whose effect also depends on the EC.

L411-413: these proportions sum to 100% but looking at Table S11, I don't think they are all proportions of the same total. Looking at Table S11, I'm struggling to understand what the denominators used for calculating the proportions are in each case, perhaps this could be clarified.

L466: Looking at Figure 4A-B, I think that 'the most photoperiod sensitive' means that the highest proportion of parental effects were determined to be photoperiod sensitive, rather than having the highest estimate for photoperiod sensitivity.

L473: I'm not sure I would describe this as interacting significantly with photoperiod without an explicit model comparison.

L500-501: It wasn't clear to me from the paragraph above that a QTLxE was detected for NODE_N on Chromosome 7.

L246: should 'environments' be 'locations'?

Manuscript GENETICS-2023-306315: Answer to reviewer and editor comments

Editor

Dear Dr. Rami:

Two experts in the field and an early career reviewer (R1) have reviewed your manuscript, and I have read it as well. The manuscript present a novel way to detect QTL by EC interactions. While your manuscript is not currently acceptable for publication in GENETICS, we would welcome a substantially revised manuscript. Both reviewers have comments and concerns to be addressed in a revised manuscript. You can read their reviews at the end of this email.

The main revisions requested are to increase clarity of writing (e.g. be very clear about the different QTL models that were fitted) and edit the manuscript to shorten some sections. The discussion seems especially long and could be more concise. We look forward to receiving your revised manuscript. Please let the editorial office know approximately how long you expect to need for revisions.

Answer. We would like to thank the editors and the reviewers for their comments and suggestions on our manuscript. It helped us to improve the quality of our work.

We prepared a revised version integrating all editors and reviewer comments with, as main modifications, a revised presentation of the QTL detection models (lines 340 – 403; **all correction position indications refer to the version with tracked changes**), a revision of the discussion concerning the limitation due to the number of environments (lines 773 - 795), and the modifications of Figures 3 and 4 (lines 523 and 594). To improve the clarity of the writing we shortened or removed many parts of the introduction, results and discussion, which compensated for the addition of reviewers' comments and still reduced the overall length of the article by 620 words. We improved the quality of the language by performing a professional proof-reading for English. The revision also allowed us to correct problems not noticed by the reviewers like the absence of heritability values for the NODE_N and NODE_L traits in the supplemental material (see new version of Table S9). We also made a small addition to Figure 2B (line 431) by mentioning that the Hafijeka variety is a Guinea Maragaritiferum type.

We hope that the modifications make our manuscript acceptable for publication in Genetics. We are looking forward for your review and decision.

Reviewer 1

R1 comment: The manuscript "Characterization of adaptation mechanisms in sorghum using a multi-reference back-cross nested association mapping design and envirotyping" presented a multi-reference sorghum back-cross nested association mapping populations composed of 24 diverse parents anchored on three West African elite lines. The sub-populations were tested for the QTLs of 8 phenotypic traits as well as the interaction between the phenotype and environmental covariables. The experiments are well-designed and performed, and the conclusions are appropriate to the results presented. I don't have any main concerns about the manuscript.

Answer: Thank you for the revision and the positive comments.

Reviewer 2

R2 comment: The paper describes a new method to detect QTLxEC interaction in BCNAM populations. In general, the method looks convincing but my main concern is the small number of environments used to demonstrate the new method. In general, the paper needs significant English correction (especially the introduction). It is hard to follow in many locations. Moreover, the length of the paper can be significantly reduced. The introduction, results and discussions all has lots of unnecessary paragraphs/statements that can be removed. It looks like that paper was copied from a thesis.

Answer: Thank you for your in-depth revision of our work and the numerous suggestions. Following them, we extended the discussion part on the number of environments (lines 773 - 795; **all correction position indications refer to the version with tracked changes**). We emphasised the fact that four environments represent a limited number of degrees of freedom to estimate sensitivity curves. We discussed an alternative strategy to extrapolate the QTL effect in untested environments through the combination of genetic analysis and crop modelling (for further details see our answers to this specific point below). We performed a revision of the English with a professional to increase the article readability. We reduced the manuscript length (620 words) by shortening or removing many parts of the introduction, results and discussion.

Abstract

R2 comment: Line 58-59: "It requires advanced crossing schemes, multi-environment characterization and relevant statistical tools" Please remove this statement as it is not fully accurate. Your approach is one excellent way to do it but it can also be done with diverse populations with or without phenotypic data (e.g. detecting selective sweeps)

Answer: We removed the statement and mentioned the existence of several approaches (lines 58-60).

R2 comment: Line 62: "The population was characterized in environments contrasting for " Please rewrite, not very clear

Answer: We rewrote the sentence like that:

"The population was phenotyped in environments characterised by differences in photoperiod, rainfall pattern, temperature levels, and soil fertility" (lines 62-64).

R2 comment: Line 67: "We mobilized ..." do you mean Utilised?

Answer: Yes, corrected (line 69).

Introduction

R2 comment: The introduction doesn't need to have subheadings.

Answer: We removed the subheadings.

R2 comment: 83-84: "and therefore genetic markers" remove

Answer: We removed that part of the sentence (line 88).

Materials and Methods

R2 comment: It is not clear if the populations were analysed independently or jointly

Answer: Because of logistic constraints, the Grinkan and Kenin-Keni populations could not be entirely phenotyped during a single season. Therefore, the lines of those sub-populations were phenotyped over two seasons (2012 and 2013). Due to minimal overlap between the genotypes phenotyped in 2012 and 2013 (maximum two crosses; 96/2109 lines in Grinkan, 23/896 lines in Kenin-Keni), we decided to analyse each Grinkan (GR2012, GR2013) and Kenin-Keni (KK2012, KK2013) sub-populations independently. Lata3 population was entirely evaluated over one season (2013). Therefore, we have a total of five sub-populations.

We emphasised that better in the text (lines 253-259).

R2 comment: The number of environments is small to have a good inference

Answer: We agree that a larger number of environments would increase the quality of the QTL effect characterization and the confidence in the statistical inference. Nevertheless, we are confident that the selection of two testing sites in the Sudanian zone (Sotuba) and at the border of the Sudano-Sahelian zone (Cinzana), as well as the different sowing dates allows the coverage of a wide spectrum of environmental conditions. This is illustrated in the important phenotypic variations observed in the different trials (see newly added supplemental Figure S5) and in the principal component analysis of the 15 covariables in the environments that show a maximum distance between the four environments (Figure S3). See modifications lines 294-295.

The characterization of large NAM populations in many environments is generally complicated. In that sense, the extent of our phenotyping is comparable to the maize EU-NAM population that was characterised in four to six environments but included less genotypes (Lehermeier et al. 2014). The maize US-NAM was characterised in five to 11 environments (Hung et al. 2012).

From a statistical point of view, we only need two environments to perform a QTLx E analysis allowing us to test the hypothesis of a variation of the QTL effect due to environmental variations. The statistical inference becomes more challenging when we attempt to link significant environmental variations of the QTL effect to specific environmental covariates (EC). The approach we implemented seeks to adjust linear functions of the EC to model the QTL allelic effect. Since we need to estimate two parameters per functions (intercept + slope), the available degrees of freedom are $(N_{env} - 2)$, which in our case is equal to two. This is the reason why we tested maximum one EC to avoid an overfit of the model.

Our strategy still allowed us to identify if the environmental QTL variation could be explained by some of the ECs, but in many cases, it was more difficult to make precise distinction between several ECs that were all showing similar degree of correlation with the QTL effect

variation. More environments would certainly help to be more discriminative about which specific EC is the most influential.

The reduced number of environments forces us to be cautious about the significance of our results but it still gives us some indications about candidate ECs influencing the QTL effect, which can be combined with other knowledge, coming for example from physiology, to refine our understanding of the QTLx E interaction. We also want to emphasise that our final QTLx EC model provides a description of the phenotypic variation through a joint estimation of multiple QTLs parental effect and sensitivity to an EC, which should be a reasonable way to maximise the information that can be extracted from our data.

Other strategies like the combination of genetic analysis and crop modelling is an alternative approach. The construction of genotype to phenotype (G2P) set-up integrating QTL expression of crop model parameters potentially allows the extrapolation of the QTL effect at the whole plant level in any environments, which relax the need for observations in many environments (Reymond et al., 2003). This strategy builds on the knowledge accumulated into the crop model to reconstruct the Gx E interaction while our QTLx EC approach is based on less complex physiological relationships derived from observed data that make the prediction dependent from the observed conditions.

We included those elements in a modified version of the discussion with an emphasis on the statistical limitations (lines 773 - 795).

Results

R2 comment: Line 379: Not all traits seem to have heritability estimation

Answer: We complemented the sentence by mentioning the heritability range for each trait (lines 442-443).

We also noticed that the supplementary material S9 did not contain the heritability results for NODE_N and NODE_L. This was corrected (see new version Table S9).

R2 comment: Line 403: "47'669, 20'343, and 20'120" I think you mean "47,669, 20,343, and 20,120"

Answer: Yes, indeed. We reformatted those numbers and the others beyond 1000, accordingly (e.g. line 469).

R2 comment: Line 404: "100 significant QTL over the five populations for eight traits, which represented 64 unique QTL" What make them unique? and are they 5 or 3 populations?

Answer: We considered that QTL positions detected for the same trait but in different sub-populations (e.g. GR2012 and Lata) represented the same unique QTL position if they were distant by less than 10 cM. For each trait, we iteratively identified unique QTL positions by searching for groups of QTLs with a maximum distance of 10 cM between the two most extreme positions starting with the positions with the smallest distance.

We rewrote the description of the unique QTL search in the manuscript (lines 407-412).

For the number of populations, as mentioned previously, five sub-populations were considered.

R2 comment: Line 405: "Consistently with the heritability estimates" this is not true, e.g. PAN. This has no relation to the heritability, it is more about the sum of effects of the major genes that can pass the significant threshold

Answer: We removed this part of the sentence. We agree that the number of QTL and variation is not directly correlated to the heritability (line 471-472).

R2 comment: Line 410: "The 100 significant QTL covered 1056 parental alleles" are you considering all significant SNPs? Please provide more details why you have >1000 alleles

Answer: At each significant QTL position, we could test $N_{parent} - 1$ allelic effects (central parent was set as reference). The total number of alleles at the significant QTL positions was. $\sum_{p=1}^{N_{sub-pop}} \sum_{i=1}^{n_{QTL}} (N_{parents(p)} - 1) = 948$. From those 948 effects, 60% of the QTL parental alleles were significant. The difference between 1056 and 948 is due to the central parent alleles that was wrongly counted in the previous version.

We clarified that in the text (lines 476-477).

R2 comment: Line 412: "and around 25% interacted with the environment" how many QTL? the same question applies for the next sentence.

Answer: 86 QTLs out of 100 had at least one parental allele that significantly interacted with the environment and 58 QTLs had at least one significant interaction with at least one EC. We added this information to the text (lines 480-481).

R2 comment: The presentation of the QTL and interaction results is very long and boring. The authors could benefit from reducing this section.

Answer: We performed a general revision of the results section to reduce it. We made small and medium modification in all sub-sections. We also decided to remove the sub-section on QTL effect and yield because those results are the most hypothetical. Contrary to the results on flowering time and plant height they are not supported by any figure and are only briefly mentioned in the discussion.

Discussion

R2 comment: The paper would benefit from adding a discussion about the effect of the small number of environments used in the present study

Answer: See our answer above concerning that question. Following your suggestion, we added two paragraphs on that topic in the discussion (lines 773 - 795).

R2 comment: Line 633: "published" should be published

Answer: Corrected.

R2 comment: 660-661: It is not clear how strong the effects are. They are representing high proportion of the phenotypic variation but the phenotypic variation could be small, which would let minor genes with small absolute effect values appear as major genes. Please investigate more on this, it would be a good idea to add a supplementary graphs showing the distribution of the phenotypic data.

Answer: We added supplementary plots and tables about the adjusted phenotype distribution (see new Figure S5). Those plots suggest that the trait variability is important. We integrated this information in the phenotypic result section (lines 438 – 441) and in the description of the QTL effects (lines 488 and 491).

R2 comment: The authors may benefit from adding a small conclusion paragraph

Answer: We added a short conclusion paragraph giving a summary of the main genomic regions detected for flowering time and plant height with their probable mode of action (lines 929 - 939).

R2 comment: Supplementary method S1: Please change "computational power" to "computational demand"

Answer: Corrected.

Reviewer 3

R3 comment. The authors present a nested association mapping population of sorghum, consisting of 3,901 (BC1F4) lines derived from 24 parents crossed to 3 recurrent parents. The population was genotyped by sequencing, providing 51,545 markers, and 8 traits were phenotyped in 4 locations across 2 years with 2 sowing time treatments and 2 phosphorus levels. The locations/treatments/crosses/years are often non-overlapping (e.g., phosphorus treatments at one location in one year for crosses with one recurrent parent) but the two sowing time treatments were performed at two locations in two years for crosses with two recurrent parents (crosses and years are semi-overlapping), which forms the core of the analysis. The authors detect QTLs and focus on interactions between QTLs and environmental covariates such as photoperiod. With a model of the sensitivity of QTLs to environmental covariates, the QTL effect can be projected beyond the measured environments. The size of the population, genetic data, and phenotyping data are impressive strengths of the study.

The interactions between QTL effects and environmental covariates shown in Figure 4 (differences in line slopes) look quite subtle. If I understand correctly (I would recommend making this part of the methods clearer and more explicit), there were three QTL models: (1) QTL model where β_{pj} in equation 2 doesn't vary across environments j (2) QTLx E model where β_{pj} varies across environments j (3) QTLx EC where β_{pj} becomes a function of sensitivity to environmental covariates (ECs) as described on L344. The last model (3) was fit for 5 ECs if the p value was lower for model (2) versus model (1) (L342).

Answer. The reviewer summary of the different models is correct. However, as we agree that the description of the QTL detection methodology could be improved, we revised the description of the models to make it more explicit. (lines 340 – 403; **all correction position indications refer to the version with tracked changes**).

The general philosophy is to first detect QTL positions and then further decompose their effect by estimating if the parental allelic effect interacts or not with the environment. For the parental allele that shows some significant interaction we further decomposed the effect by estimating allelic sensitivity to specific environmental covariates.

R3 comment. I'm not clear on the method used to determine if the EC interactions were significant (L348). In general, I don't think an explicit method was used for model comparison (e.g., LR test) beyond the comparison of p values, which makes it hard to interpret the (statistical) significance of the subtle changes in slope shown in Figure 4. The effect size also suggests caution about the agronomic significance of environmental sensitivity relative to the main effect of the QTLs.

Answer. All the tests about significance of the QTL effects, their mode of action, and the significance of the allelic sensitivity to the EC were realised using a Wald test. The significance of the effect was estimated by comparing the test statistics under the null hypothesis (absence of effect) which is supposed to follow a chi-squared distribution with the degree of freedom equal to the number of tested effects (e.g. the number of parental alleles, or 1 for the slope term).

According to us, this is a standard procedure used to evaluate the significance of fixed effect in mixed models. This strategy was already used by several authors to specifically evaluate the sensitivity of a QTL to EC . For example, Boer et al. (2007) or Millet et al. (2016).

Maybe the reviewer's doubt was due to the fact that the explicit mention of the Wald test was minimal in the description of the methodology. We corrected that (lines 349 – 352, 386, 399).

Specific comments:

R3 comment. Figure 3: the use of a dot and a circle is a little confusing because some small circles can look like dots. Perhaps a cross could be used instead of a dot. The parent names are also difficult to read, I suggest adjusting them so that they begin at the tick mark.

Answer. The figure was corrected according to the reviewer's suggestion (line 523).

R3 comment. Figure 4A-F: I think the only parental effects shown are those that have lower p values under the QTLxE model. It would be nice to also display the other parental effects (i.e., those that have no significant effect relative to the recurrent parent and those that don't show evidence of QTLxE. I think the model of environmental sensitivity comes from four environments (two locations and two sowing dates). Ideally, we could also see the trait values in these four environments or, at least, note the EC value at these locations/sowing dates on the X axis. I'm also slightly confused about the EC range because Table S4 suggests the maximum range for measured photoperiod should be 11.9-12.3.

Answer. We made a new version of Figure 4A-F (line 594) where the effects of all parents are plotted. For that purpose, we recalculated the final model to estimate the sensitivity of all parents. Following the reviewer suggestion, we also plotted the EC values on the x axis. The intersection between the EC vertical lines and the QTL effect sensitivity slope gives the QTL additive values in each environment.

When we modified the figure, we noticed some issues in the construction and the interpretation of the sensitivity curves. In the initial version, the intention was to represent the trait value with the deviation due to the QTL allelic effect. Therefore, we plotted the sum of the cross-specific (donor parent) intercept and the QTL deviation across environments. In that way, most of the curve shape is given by the average cross (donor parent) behaviour rather than the QTL effect. Therefore, we revised the figure by plotting only the QTL effect term, which includes the intercept term ($\hat{\beta}_p$) representing the parent allele main additive effect across environments and the allelic sensitivity ($EC_e * \hat{S}_p$) representing the variation of the additive effect given the change in EC.

We are aware that, in most of the presented cases, the EC variation looks small. However, those small variations can have an important effect at the phenotypic level and strongly influence the QTL effects. For example, for the largest GR2012 population, we could notice an average difference of 18 days in the flowering between plants grown under an environment with 12.1 hours of photoperiod and the one that experienced 12.4 hours. Similarly, the plant height of GR2013 plants measured in two environments differentiated by around 2 degrees of temperature range (SB1: 8.4, CZ2: 10.3) was on average 59 cm. Those important phenotype changes could be linked to QTL effect variations. For example, the effect of parent SC56614 at QTL_FL_6_36 varies by around three days over 0.3 hours of photoperiod. The effect of parent Hafijeka at QTL_PH_7_75 potentially varies by around 20 cm over 2 degrees of temperature range (lines 556-558).

We added the new figure and revised the caption (lines 596-600) as well as the results description (lines 539-545, 587-588) to reflect those changes.

R3 comment: Figure 4G: This is very difficult to understand without further explanation.

Answer: We revised the description of Figure 4G (lines 607 - 614). We also made small modification on the figure like the addition of several steepness for the photoperiod sensitivity to make it more explicit. We noticed that the figure is only mentioned a single time at the beginning, so we added other mention of the figure in the text, which should also make figure 4G more understandable (lines 835 and 870).

R3 comment: Figure 5: Since these are split by QTL effects, isn't the quantity of interest the deviation from the reference genotype, whose effect also depends on the EC.

Answer: Indeed, the effects plotted on Figure 5 represent the deviation with respect to the reference genotype (central parent) in the "reference" environments, which was the one with the lowest EC value. We added this information to the description of Figure 5 results and in its caption (lines 648 – 650 and 666 - 667).

R3 comment: L411-413: these proportions sum to 100% but looking at Table S11, I don't think they are all proportions of the same total. Looking at Table S11, I'm struggling to understand what the denominators used for calculating the proportions are in each case, perhaps this could be clarified.

Answer: In each combination of sub-population (e.g. GR2012) by trait (e.g. FLAG), the initial QTL scan, gave us a number of significant QTL positions. Each of those QTL positions is defined by $N_{\text{parent}} - 1$ (central parent set as reference) alleles. The total number of parental alleles at the significant QTL positions is the denominator for each proportion table. Then, for each of those parental alleles, we could further determine if they are significant or non-significant. If they are significant, if they significantly interact with the environment. And if they interact with the environment, if there is at least one environmental covariate with which they interact. Therefore, the different tables contain the total number, respectively the proportion of significant parental alleles, of parental alleles with a significant QTLxE effect, and with a significant QTLxEC effect. The tables describe the degree of (parental) allelic significance, allelic interaction with the environment, and allelic interaction with identified ECs.

We clarified this element of the supplementary material by adding some explanations and by reducing the table. We added the total number of QTL parental alleles in each configuration and kept only the proportions (see new version of table S11).

R3 comment: L466: Looking at Figure 4A-B, I think that 'the most photoperiod sensitive' means that the highest proportion of parental effects were determined to be photoperiod sensitive, rather than having the highest estimate for photoperiod sensitivity.

Answer: We re-wrote this part of the text like that:

"QTL_FL_6_3 (*Ma6*) was the QTL with the highest proportion of alleles influenced by the photoperiod" (lines 537-538).

R3 comment: L473: I'm not sure I would describe this as interacting significantly with photoperiod without an explicit model comparison.

Answer. Concerning the estimation of the QTLxEC significance, we clarified that point in our answer to the general comments. Maybe part of the reviewer's doubt comes from the fact that the significance test was too briefly described in the material and method? We added more details and clarification in our revision of the model description (lines 377- 403).

R3 comment. L500-501: It wasn't clear to me from the paragraph above that a QTLxE was detected for NODE_N on Chromosome 7.

Answer. Indeed, there is no QTL for NODE_N on chromosome 7. The QTLs for NODE_N are located on chromosomes 3 and 6. In the pointed sentence we try to find an explanation for the significant interaction between the PH QTL on chromosome 7 and the photoperiod sensitivity. The corresponding QTL for NODE_L is not photoperiod sensitive. Therefore, we made the hypothesis that the photoperiod sensitivity of the QTL on chromosome 7 could be due to the PH part that is controlled by the NODE_N component but via QTL on other chromosomes, in that case chromosome 6.

We rephrased the sentence to clarify that point (lines 581-583).

R3 comment. L246: should 'environments' be 'locations'?

Answer. Indeed, it is locations. We corrected it (line 265).

References

- Boer, M. P., Wright, D., Feng, L., Podlich, D. W., Luo, L., Cooper, M., & Van Eeuwijk, F. A. (2007). A mixed-model quantitative trait loci (QTL) analysis for multiple-environment trial data using environmental covariables for QTL-by-environment interactions, with an example in maize. *Genetics*, *177*(3), 1801-1813.
- Hung, H. Y., Browne, C., Guill, K., Coles, N., Eller, M., Garcia, A., ... & Holland, J. B. (2012). The relationship between parental genetic or phenotypic divergence and progeny variation in the maize nested association mapping population. *Heredity*, *108*(5), 490-499.
- Lehermeier, C., Krämer, N., Bauer, E., Bauland, C., Camisan, C., Campo, L., ... & Schön, C. C. (2014). Usefulness of multiparental populations of maize (*Zea mays* L.) for genome-based prediction. *Genetics*, *198*(1), 3-16.
- Millet, E. J., Welcker, C., Kruijer, W., Negro, S., Coupel-Ledru, A., Nicolas, S. D., ... & Tardieu, F. (2016). Genome-wide analysis of yield in Europe: allelic effects vary with drought and heat scenarios. *Plant Physiology*, *172*(2), 749-764.
- Reymond, M., Muller, B., Leonardi, A., Charcosset, A., & Tardieu, F. (2003). Combining quantitative trait loci analysis and an ecophysiological model to analyze the genetic variability of the responses of maize leaf growth to temperature and water deficit. *Plant physiology*, *131*(2), 664-675.

November 18, 2023

GENETICS-2023-306571

Characterization of adaptation mechanisms in sorghum using a multi-reference back-cross nested association mapping design and envirotyping

Dear Dr. Rami:

Two experts in the field have reviewed your manuscript, and I have read it as well. I am pleased to inform you that, with minor revisions, it is potentially suitable for publication in GENETICS. The reviewers have comments and concerns that need to be addressed in a revised manuscript. You can read their reviews at the end of this email.

It is most important that you address the following in your resubmission: clarify the number parental allelic effects and the classification of QTL having a significant interactions.

We look forward to receiving your revised manuscript. Please let the editorial office know approximately how long you expect to need for revisions.

Upon resubmission, please include:

1. A clean version of your manuscript;
2. A marked version of your manuscript in which you highlight significant revisions carried out in response to the major points raised by the editor/reviewers (track changes is acceptable if preferred);
3. A detailed response to the editor's/reviewers' comments and to the concerns listed above. Please reference line numbers in this response to aid the editors.

Additionally, please ensure that your resubmission is formatted for GENETICS.

<https://academic.oup.com/genetics/pages/general-instructions>

Follow this link to submit the revised manuscript: Link Not Available

Sincerely,

Hans Daetwyler
Associate Editor
GENETICS

Approved by:
Mario Calus
Senior Editor
GENETICS

Reviewer #2 (Comments for the Authors (Required)):

Thanks for your efforts in reviewing the manuscript. The updated version is much improved and answered most of my concerns. I only left with one comment regarding the "QTLxEC extend"

In line 401, you claimed that you have Npar-1 allelic effects which I strongly disagree with. You genotyped your population with bi-allelic SNPs so regardless of the number of parents you have, you only end with 2 alleles with stabled modelled effects given that your method assumes only additive inheritance. I would appreciate if you can revise this throughout the manuscript.

Reviewer #3 (Comments for the Authors (Required)):

There have been substantiative positive revisions to the manuscript. In particular, I think that the updated version of Figure 4 is much clearer in showing the interactions of interest. The description of the model fitting procedure is also much easier to understand. The revisions have also made the biological significance of the environmental interactions clearer, which is an important component of the study, along with the strengths identified previously.

The classification of QTL as having significant interactions or not is an outstanding area of confusion for me. Equation 3 includes a main effect term and an interaction term and the text below says that a Wald test was used to determine whether both/either were significant. However, the text then says that the p values for the interaction term and the main effect term were compared to see if there is more support for a main effect or an interaction. Unless I have misunderstood, I think this text means that QTL were considered to have a significant interaction (e.g., L548 and solid versus dashed lines in Figure 4) only if the p value for the interaction was lower than that of the main effect in this model. I would think it makes more sense to use the significance of the interaction term to decide whether the interaction is significant, rather than relying on a comparison with the main effect (i.e., a significant interaction shouldn't be hidden by a strong main effect, which may have occurred for CSM388 in Figure 4D). If the question is whether the more complex model (including interactions) is a significantly better fit than the null model of only main effects, then I would use an F test or a LR test to compare the models. I would recommend that this issue is further addressed because the classification of QTL as having significant interactions or not is an important component of the study.

Figure 3: I think that the QTL effect size plotted here is both the parent allele main additive effect across environments and the allelic sensitivity, as in Figure 4, is that correct?

Manuscript GENETICS-2023-306571: Answer to reviewer and editor comments

Editor

Dear Dr. Rami:

Two experts in the field have reviewed your manuscript, and I have read it as well. I am pleased to inform you that, with minor revisions, it is potentially suitable for publication in GENETICS. The reviewers have comments and concerns that need to be addressed in a revised manuscript. You can read their reviews at the end of this email.

It is most important that you address the following in your resubmission: clarify the number parental allelic effects and the classification of QTL having a significant interactions.

We look forward to receiving your revised manuscript. Please let the editorial office know approximately how long you expect to need for revisions.

Answer: Dear Editors and reviewers. We would like to thank you for the in-depth attention you gave to our manuscript. The points you raised were highly relevant and allowed us to further improve the quality of our work.

Following reviewer 3 comments, we decided to revise the statistical methodology implemented to estimate the QTL by environment effect and the sensitivity of the QTL allelic effect to specific environmental covariate (lines 339-369; **all the indications refer to the tracked changes version**). Those modifications should strengthen the formalism of our methodology and facilitate the understanding of the procedure. The revision of the computation procedure made us notice a mistake in our code that was corrected. Therefore, we recalculated all the QTL by environment effect and QTL by environmental covariate effects. These modifications did not significantly modify the achieved results (lines 428-430, 507-509; Table 3 (line 1106) number of significant parental effects; supplementary tables S11 and S12). We also adjusted Figure 3 which was based on the old procedure (line 470; for further detail see answer to reviewer 3) and Figure 5 with the updated results (line 563).

Concerning the remark about the number of allelic effects at the QTL position made by reviewer 2, we agree that the SNP marker scores rely on two allelic classes, but we still consider that the combination of the marker score with the design allow the assumption of a multi-allelic model with one different allele associated to each parent. According to us, such a model is the most pertinent to describe our data and it is also justified from a biological point of view. We provided several elements of justification for this choice in our answer to reviewer 2 and reported those in the manuscript (lines 298 – 308 and 426). We emphasized the inferential nature of our model concerning the number of alleles. We hope that those extra elements of justification and modifications help to solve this question.

We also benefited from this revision to update the data availability and acknowledgment sections (lines 753-759 and 764-765). We hope that those modifications make our work suitable for a publication in Genetics.

Reviewer 2

R2 comment: Thanks for your efforts in reviewing the manuscript. The updated version is much improved and answered most of my concerns. I only left with one comment regarding the "QTLxEC extend"

In line 401, you claimed that you have Npar-1 allelic effects which I strongly disagree with. You genotyped your population with bi-allelic SNPs so regardless of the number of parents you have, you only end with 2 alleles with stabled modelled effects given that your method assumes only additive inheritance. I would appreciate if you can revise this throughout the manuscript.

Answer: The population was indeed genotyped with bi-allelic SNP markers, which relies on the existence of two differences in the base pair at the marker. However, we consider that the inference of more than two alleles based on the bi-allelic marker score information in multi-parent populations (MPPs) is supported by several biological considerations. First, we know that complex traits are controlled by many QTL/genes with difference of allelic frequencies in different cross including the absence of segregation or the fixation (Xu 1998, Holland 2007). Second, even if the marker density is high, the probability of a perfect linkage between the marker and the QTL/gene position is low and the linkage disequilibrium between the marker and the QTL/gene can vary between the different crosses (Bauer et al. 2013). Therefore, the combination of cross-specific genetic make-up with different allele frequencies and/or difference in linkage disequilibrium can produce different QTL effects similar to a multi-allelic configuration. Third, the allelic configuration at the marker position could also be multi-allelic (Jannink and Wu 2003; Giraud et al. 2014). The bi-allelic marker score metric will only be partially correlated to the true allelic dosage, but it will still result in differences of effects that are best captured by a multi-allelic (parental) QTL effect.

The marker score information can be used in different ways to infer the genetic relatedness between genotypes at the marker position. From an identical by state (IBS) point of view, we can assume two alleles that correspond to the two marker scores. However, we can also combine the marker score with other information like the crossing scheme, the neighbouring markers scores, and/or the assumed recombination pattern to infer identical by descent (IBD) relatedness between the genotypes. Such an approach allows the definition of more than two allelic effects, which is one of the main interests of the MPP designs. For example, in the NAM setting, we can consider each cross separately to infer parental allele origin and consider that at the i th marker, genotypes with the same marker score as the donor parent inherit their allele from this parent while parent with the same score as the central parent inherit their allele from the central parent. This method allows the definition of N_parent alleles. We applied such a strategy, which is a common approach in MPP QTL analysis that has been implemented in several articles (Li et al. 2011, Bardol et al. 2013, Garin et al. 2017). From a statistical point of view, in a NAM design since each donor parent is used in one cross, the parental model cannot be distinguished from a model assuming cross-specific effects with an additive parental modulated by the cross genetic background (Jansen et al. 2003). Such a definition of the QTL term can model the variability of effect due to multiple alleles, difference of linkage, and/or QTL/gene segregation at the QTL position.

The assumption of a parental multi-allelic system is also useful, because from a statistical point of view, it represents the most exhaustive model which could capture the largest diversity of effects at the QTL. From a breeding point of view, it allows to estimate the effect of a specific parent and trace back the origin of the effect to an identified line, which is crucial information for breeders for the design of new crosses (Christopher et al., 2021).

Based on the elements above, we still consider that the parental model used so far assuming N_{parent} alleles is justified. Using a bi-allelic model is an alternative but it could represent a loss of information. Therefore, to take into consideration the reviewer comment, we revised the justification supporting the use of a multi-allelic parental model and we better emphasized the fact that the parental allelic effects represent an assumption (lines 298 – 308 and 426).

Reviewer 3

Reviewer comment: There have been substantive positive revisions to the manuscript. In particular, I think that the updated version of Figure 4 is much clearer in showing the interactions of interest. The description of the model fitting procedure is also much easier to understand. The revisions have also made the biological significance of the environmental interactions clearer, which is an important component of the study, along with the strengths identified previously.

The classification of QTL as having significant interactions or not is an outstanding area of confusion for me. Equation 3 includes a main effect term and an interaction term and the text below says that a Wald test was used to determine whether both/either were significant. However, the text then says that the p values for the interaction term and the main effect term were compared to see if there is more support for a main effect or an interaction. Unless I have misunderstood, I think this text means that QTL were considered to have a significant interaction (e.g., L548 and solid versus dashed lines in Figure 4) only if the p value for the interaction was lower than that of the main effect in this model. I would think it makes more sense to use the significance of the interaction term to decide whether the interaction is significant, rather than relying on a comparison with the main effect (i.e., a significant interaction shouldn't be hidden by a strong main effect, which may have occurred for CSM388 in Figure 4D). If the question is whether the more complex model (including interactions) is a significantly better fit than the null model of only main effects, then I would use an F test or a LR test to compare the models. I would recommend that this issue is further addressed because the classification of QTL as having significant interactions or not is an important component of the study.

Answer: Thank you for this comment that gives us the opportunity to further clarify our methodology description because your comment made us notice that it is not understood as it should be. We also noticed that the way we tested for the significance of the QTL by environment interaction (QEI) is not straightforward, which probably explains the confusion about this part of the manuscript. Therefore, we decided to revise the methodology used to determine the significance of the QEI term (lines 339-369). The revised model sequence is the following:

- 1) Detection of QTL positions by performing a genome scan (simple and composite interval mapping) using model 2 (unchanged)

$$y_{icj} = E_j + C_{cj} + x_{iqp} * \beta_{pj} + GE_{icj} + e_{icj} \quad (2)$$

- 2) For each QTL, determine if the parental alleles effects composing the overall QTL effect have significant interaction with the environment. For that purpose, we decomposed the parental QTL effects into a main effect component (α_p) and a QEI component (β_{pj}) that were estimated simultaneously, which gives the following multi-QTL model

$$\underline{y_{icj}} = E_j + C_{cj} + \sum_{q=1}^{n_{QTL}} x_{iqp} * (\alpha_p + \beta_{pj}) + \underline{GE_{icj}} + \underline{e_{icj}} \quad (3)$$

We tested for the significance of the α_p and β_{pj} using the Wald test, which is a standard procedure to evaluate fixed effect in mixed models and is an equivalent to the F-test in this context. Those tests allowed us to determine if the parental effect was significant (α_p or β_{pj} significant), and if the parental allele interacted with the environment (β_{pj} significant).

- 3) For the parents with a significant QEI (pxE), we replaced the β_{pj} with $EC_j * S_p + l_{p\epsilon}$ to test the environmental sensitivity of the QTL allele with respect to the environmental covariate EC_j (e.g. humidity) that represent the EC value in environment j associated with the sensitivity term S_p and the residual effect $l_{p\epsilon}$. The fitted model was the following:

$$\underline{y_{icj}} = E_j + C_{cj} + \sum_{q=1}^{n_{QTL}} x_{iqp} * (\alpha_p + \beta_{pj}) + x_{iqpxE}(\alpha_p + EC_j * S_p + l_{p\epsilon}) + \underline{GE_{icj}} + \underline{e_{icj}} \quad (4)$$

We estimated this model for the five most significant EC selected previously and determined the significance of the sensitivity term using again the Wald test. Therefore, in both cases, the QEI and S_p were estimated by estimating those effects conditionally on the main effect term. Their significance was evaluated by looking at the Wald test statistics associated with those terms only. Such a procedure is comparable to the approach implemented in Boers et al. (2007). We recalculated the results given those changes. We hope that those modifications made the procedure easier to understand and clarify your doubt.

Reviewer comment: Figure 3: I think that the QTL effect size plotted here is both the parent allele main additive effect across environments and the allelic sensitivity, as in Figure 4, is that correct?

Answer: Given the modification of the statistical methodology we slightly revised Figure 3 (line 470). In the previous version, the main parental effects and the QTL by environment effects were estimated in different models, which makes their comparison indirect. The effects estimated with the updated version of model 3 represent the deviations with respect to the main effect. According to us, the effect estimated using model 2 representing the estimated parental allelic effect in each environment allow a better visualisation of the QTLxE effect range.

References

- Bauer, E., Falque, M., Walter, H., Bauland, C., Camisan, C., Campo, L., ... & Schön, C. C. (2013). Intraspecific variation of recombination rate in maize. *Genome biology*, *14*, 1-17.
- Bardol, N., Ventelon, M., Mangin, B., Jasson, S., Loywick, V., Couton, F., ... & Moreau, L. (2013). Combined linkage and linkage disequilibrium QTL mapping in multiple families of maize (*Zea mays* L.) line crosses highlights complementarities between models based on parental haplotype and single locus polymorphism. *Theoretical and applied genetics*, *126*(11), 2717-2736.
- Boer, M. P., Wright, D., Feng, L., Podlich, D. W., Luo, L., Cooper, M., & Van Eeuwijk, F. A. (2007). A mixed-model quantitative trait loci (QTL) analysis for multiple-environment trial data using environmental covariables for QTL-by-environment interactions, with an example in maize. *Genetics*, *177*(3), 1801-1813.
- Christopher, M., Paccapelo, V., Kelly, A., Macdonald, B., Hickey, L., Richard, C., Verbyla, A., Chenu, K., Borrell, A., Amin, A., Christopher, J. (2021). QTL identified for stay-green in a multi-reference nested association mapping population of wheat exhibit context dependent expression and parent-specific alleles. *Field Crops Res.* *270*, 108181.
- Garin, V., Wimmer, V., Mezouk, S., Malosetti, M., & Van Eeuwijk, F. A. (2017). How do the type of QTL effect and the form of the residual term influence QTL detection in multi-parent populations? A case study in the maize EU-NAM population. *Theoretical and Applied Genetics*, *130*(8), 1753-1764.
- Giraud, H., Lehermeier, C., Bauer, E., Falque, M., Segura, V., Bauland, C., ... & Moreau, L. (2014). Linkage disequilibrium with linkage analysis of multiline crosses reveals different multiallelic QTL for hybrid performance in the flint and dent heterotic groups of maize. *Genetics*, *198*(4), 1717-1734.
- Holland, J. B. (2007). Genetic architecture of complex traits in plants. *Current opinion in plant biology*, *10*(2), 156-161.
- Jannink, J. L., & Wu, X. L. (2003). Estimating allelic number and identity in state of QTLs in interconnected families. *Genetics Research*, *81*(2), 133-144.
- Jansen, R. C., Jannink, J. L., & Beavis, W. D. (2003). Mapping quantitative trait loci in plant breeding populations: use of parental haplotype sharing. *Crop Science*, *43*(3), 829-834.
- Li, H., Bradbury, P., Ersoz, E., Buckler, E. S., & Wang, J. (2011). Joint QTL linkage mapping for multiple-cross mating design sharing one common parent. *PloS one*, *6*(3), e17573.
- Xu, S. (1998). Mapping quantitative trait loci using multiple families of line crosses. *Genetics*, *148*(1), 517-524.

December 20, 2023

RE: GENETICS-2023-306737

Dr. Jean-François Rami
CIRAD
BIOS
TA A-96/03 Av Agropolis
Montpellier Cedex 5 34398
France

Dear Dr. Rami:

Congratulations! We are delighted to inform you that your manuscript entitled "Characterization of adaptation mechanisms in sorghum using a multi-reference back-cross nested association mapping design and envirotyping" is acceptable for publication in GENETICS. Many thanks for submitting your research to the journal.

To Proceed to Production:

Add oupsupport@scipris.com and genetics.oup@novatechset.com (or the domains @scipris.com and @novatechset.com) to your email program's "safe senders" list. You will be contacted by both at various points during the production process.

1. Format your article according to GENETICS style, as discussed at <https://academic.oup.com/genetics/pages/general-instructions>. Ensure that you comply with data and community resource citation guidelines (<https://academic.oup.com/genetics/pages/general-instructions#Data-Policy>).
2. Upload your final files at <https://genetics.msubmit.net>. The GSA Journals use SciPris to manage article licensing and payment. If you do not have a SciPris account, you will receive an email from no-reply@scipris.com to sign up to use Oxford University Press' author portal. After logging in, follow the online instructions to sign your licence and arrange any payment due.
3. Your currently-accepted manuscript (unedited, as submitted, reviewed, and accepted) will be published at GENETICS and deposited into PubMed as an Advance Access article. Notify sourcefiles@thegsajournals.org before signing your license if you do not wish to publish your article via Advance Access.
4. We invite you to submit an original color figure related to your paper for consideration as cover art. Please email your submission to the editorial office or upload it with your final files. You can submit a small-sized image for evaluation, and if selected, the final image must be a TIFF file 2513px wide by 3263px high (8.375 by 10.875 inches; resolution of 600ppi). Please avoid graphs and small type.

If you have any questions or encounter any problems while uploading your accepted manuscript files, please email the editorial office at sourcefiles@thegsajournals.org.

Sincerely,

Hans Daetwyler
Associate Editor
GENETICS

Approved by:
Mario Calus
Senior Editor
GENETICS

Review comments (if applicable):

I really appreciated the engagement and willingness to reanalyse the dataset based in the reviewer comments. Thank you!
Hans